evolution, environmental science

phytoplankton, climate change, experimental evolution, phenotypic plasticity

**Authors for correspondence:**
C.-E. Schaum
e-mail: elisa.schaum@uni-hamburg.de
G. Yvon-Durocher
e-mail: g.yvon-durocher@exeter.ac.uk

# Evolution of thermal tolerance and phenotypic plasticity under rapid and slow temperature fluctuations

C.-E. Schaum[1,3], A. Buckling[1], N. Smirnoff[2] and G. Yvon-Durocher[1]

[1]University of Exeter, Penryn Campus, Penryn, Cornwall TR10 9EZ, UK
[2]Biosciences, College of Life and Environmental Sciences, Geoffrey Pope Building University of Exeter, Exeter EX4 4QD, UK
[3]Centre for Earth Systems and Sustainability (CEN)/ Institute for Marine Ecosystems and Fishery Science (IMF), Hamburg University, 22767 Hamburg, Germany

C-ES, 0000-0001-6949-7367

Global warming is associated with an increase in sea surface temperature and its variability. The consequences of evolving in variable, fluctuating environments are explored by a large body of theory: when populations evolve in fluctuating environments the frequency of fluctuations determines the shapes of tolerance curves (indicative of habitats that organisms can inhabit) and trait reaction norms (the phenotypes that organisms display across these environments). Despite this well-established theoretical backbone, predicting how trait and tolerance curves will evolve in organisms at the foundation of marine ecosystems remains a challenge. Here, we used a globally distributed phytoplankton, *Thalassiosira pseudonana*, and show that fluctuations in temperature on scales of 3–4 generations rapidly selected for populations with enhanced trait plasticity and elevated thermal tolerance. Fluctuations spanning 30–40 generations selected for the formation of two stable, genetically and physiologically distinct populations, one evolving high trait plasticity and enhanced thermal tolerance, and the other, akin to samples evolved under constant warming, with lower trait plasticity and a smaller increase in thermal tolerance.

## 1. Introduction

In a warming world set to see an unprecedented increase in thermal variability [1], ectotherm fitness will be directly intertwined with how fast, and by how much temperature increases. Growth rates and similar fitness proxies of ectotherms exhibit unimodal (hump-shaped) responses to temperature [2,3]. These 'thermal tolerance curves' serve as an indicator of the range of thermal conditions in which organisms can persist [4] and can serve as a baseline for predicting future distributions of species under global warming. Thermal tolerance curves are not fixed. Selection experiments using diverse taxa from insects to micro-algae and bacteria, indicate that key parameters of thermal tolerance, such as the optimum (i.e. the temperature at which growth is maximal) and the maxima (i.e. the upper limit of thermal tolerance) can rapidly shift as organisms evolve in response to changes in their thermal environment [5–7]. Rapid evolution of increased thermal tolerance could be essential in facilitating the persistence of species under unprecedented rates of environmental change.

Much of our understanding about the rapid evolution of thermal tolerance comes from experimental studies that consider responses to constant warming. Thermal environments in nature are anything but constant, and global warming is expected to further amplify natural temperature variability. Scientific understanding about the impacts of thermal variability on the rapid evolution of thermal tolerance is exceptionally limited. What we do understand comes

mostly from theory or comparative studies where species' thermal tolerance are linked to the thermal variability of the environments they inhabit. Theory and data typically agree that species from higher latitudes that experience greater thermal variability tend to have broader thermal tolerance curves than their counterparts from thermally stable environments in low latitudes [8,9], which tend to be thermal specialists with narrow tolerance ranges. The contrast between the evolution of thermal generalists and specialists is thought to manifest as a trade-off between the breadth and height of the tolerance curve. Adaptation to variable environments leads to broad but flat tolerance curves characteristic of thermal generalists, whist evolution of specialists in stable environments leads to steep, narrow tolerance curves. However, making causative inference about the links between environmental variability and the shape of tolerance curves from comparative studies is very challenging because of the multitude of other environmental factors that covary with thermal variability along latitudinal gradients. Experimental studies that explore the impact of thermal fluctuations on the evolution of thermal tolerance, while controlling for confounding factors could help significantly with building knowledge to support inference on the links between environmental variability and tolerance curve evolution.

A key pathway toward understanding how environmental variability shapes the evolution of tolerance curves could come from understanding how the traits that underlly the impacts of temperature change on fitness evolve under different scenarios of thermal fluctuations. Metabolic traits, such as rates of respiration and photosynthesis, are known to respond significantly to changes temperature and play important role in shaping the limits of thermal tolerance and the capacity for the evolution of elevated thermal tolerance [10]. The dynamic response of a metabolic trait to temperature change can be considered a 'reaction norm'. Within the linear part of the reaction norm, the steepness of the slope is indicative of the strength of an organisms' plastic response (i.e. the ability of the same genotype to express new phenotypes in response to environmental change [11,12]). Steeper reaction norms indicate higher phenotypic plasticity. Theory describes the links between plasticity and evolutionary potential [13–16], and the evolution of plasticity itself across stable and fluctuating environments [16–20]. The latter state that greater plasticity should evolve in predictably fluctuating environments due to the fitness advantages associated with phenotypic plasticity in a variable environment. Theory by Lande [21] formally connects reaction norms of trait plasticity to tolerance curves in fluctuating environments and demonstrates how greater environmental variability leads to higher phenotypic plasticity and broader tolerance curves. Lande's theory of plasticity and environmental tolerance [21], also captures the often hypothesized trade-off between the breadth and height of the tolerance curve by invoking fitness costs to plasticity that preclude the evolution of tolerance curves that are both high and broad. However, empirical validation of the links between environmental variability, trait plasticity and the evolution of tolerance curves is currently severely lacking.

Here we address these knowledge gaps by leveraging a long-term selection experiment [22] with the model marine diatom, *Thalassiosira pseudonana* which has evolved for 300 generations in fluctuating environments that differed in the *frequencies* of fluctuations (3–4 or 30–40 generations). The fluctuations switched between 22°C and 32°C, and were run alongside stable environmental conditions at 22°C and 32°C. These temperatures represent control conditions at 22°C and warming at supra-optimal temperature at 32°C. We quantified the magnitudes of the evolutionary responses and determined whether the evolution of the thermal tolerance curve and magnitude phenotypic plasticity for each selection lines varied between stable environments and those that fluctuated on shorter versus longer frequencies. We also resequenced the genomes of the ancestor and all selections to determine whether changes in in fitness and phenotypic traits were also reflected in patterns of molecular evolution.

## 2. Material and methods

This is a methods summary. Detailed methods that allow for reproduction or re-analysis of the experiment can be found in the electronic supplementary material.

### (a) Experimental design

A sequenced strain of *Thalassiosira pseudonana* (CCMP 1335) [23] was obtained from the CCAP culture collection in November 2014. The stock culture was made clonal by serial dilution. The selection regimes for 300 generations (see electronic supplementary material, figure S8) were: (i) the benign control temperature at 22°C, which was the temperature that the isolate had been maintained at in the culture collection; (ii) constant, extreme warming at 32°C, as pilot experiments with the ancestor had revealed that growth rates peaked at 28°C and were reduced above 35°C; (iii) a 'short' fluctuating treatment (hereafter 'FS'), where temperature switched between 22°C to 32°C every 3–4 generations; and (iv) a second, 'long' fluctuating treatment (hereafter 'FL'), where temperature switched between 22°C and 32°C every ~40 generations. Each treatment was replicated 6 times. In this specific set-up, the geometric mean environment between FS and FL remains the same, and they are also equal in their predictability and amplitudes. This set-up was chosen specifically to test for the effect of differences in frequencies of fluctuations. Throughout the experiment, lineages were grown in grown in f/2 medium (Guillard's medium for diatoms [24]) with artificial seawater, under a 12 : 12 light/dark cycle. Salinity was maintained at 32 (i.e. 32 g NaCl $l^{-1}$ in 39.5 g $l^{-1}$ artificial seawater reagents) and light intensity was at 100 µmol quanta $m^{-2} s^{-1}$. Lineages were maintained in semi-continuous batch culture, transferred during the exponential phase. After 300 generations of selection in their respective environments, all samples were used for a full reciprocal transplant assay.

### (b) Growth rate trajectories

Abundance, size and fluorescence of cells were determined at each transfer using an Accuri C6 flow cytometer (BD Scientific).

At the beginning and at the end of each transfer *T. pseudonana* cells from each selection environment were counted on the flow cytometer as described above and used to estimate specific growth rates ($u$ $d^{-1}$).

### (c) Thermal tolerance curves of growth

To characterize thermal tolerance curves of growth, an inoculum of 100 cells per ml from the middle of the logarithmic phase of growth was transferred into fresh media at 15°C, 20°C, 25°C, 30°C, 32°C, 35°C and 40°C, and cell count was then determined daily on an BD accuri C6 flow cytometer.

### (d) Photochemistry

We characterized a suite of photochemical parameters in the ancestor and each of the evolved lineages using fast repetition rate fluorometry

(FastPro8, FRRf3, Fast Ocean System Chelsea Technology Group). Measurements were taken at 22°C and 32°C. Photochemical traits were measured in response to rapid flashes at increasing light intensities from 0 to 1600 µmol m$^{-2}$ s$^{-1}$. Flash frequency and rate followed standard protocols for phytoplankton [25], with 100 flashes of 1.1 µs at 1 µs intervals. Peak emission wavelengths of the LEDs used for excitations were at 450 nm, 530 nm and 624 nm.

### (e) Plasticity in photosynthesis and related photochemical traits

Rates of gross photosynthesis were measured in the ancestor and all evolved lineages after 300 generations of selection when in the middle of the logarithmic phase of population growth using a Clark-type oxygen electrode. Net photosynthesis (NP) was measured as $O_2$ evolution at increasing light intensities up to 2000 µmol$^{-1}$ m$^{-2}$ s$^{-1}$. The maximum rate of light-saturated photosynthesis was determined by fitting the NP data to a dynamic model of photoinhibition via nonlinear least squares regression using the methods outlined in [26]. To estimate plastic responses from the gross photosynthesis data, expressed in units of µmolO$_2$ per cell and hour, were transformed to µg C per µg C following [27]. The µg C per µg C data were plotted as a function of assay temperature. The steepness of the resulting slopes was used as our measure of plasticity, with steeper slopes indicating higher plasticity (see also electronic supplementary material, figure S11 for a flow chart). Comparing slopes of the evolved lines allows us to investigate whether different environments yield populations with higher or lower plasticity. Comparing our samples to the ancestor (or even the 22°C evolved samples to control for evolution in response to laboratory settings) allows us to investigate how these plastic responses evolved. Plastic responses were measured on a population level, and therefore, responses could either be due to higher plasticity on a cellular level, or increased metabolic diversity within one sample. While ideally one should measure plastic responses on the level of single clones isolated from the evolved samples, this is not possible for the methods chosen here. We did, however, re-clonalize all evolved samples (96 clones per sample; see electronic supplementary material, figure S6) and found that within replicate variation for growth was extremely low.

### (f) DNA extraction for whole genome re-sequencing

DNA was extracted following a standard cTAB protocol [28]. Samples were processed at the Exeter Sequencing Services (University of Exeter, UK). The resulting reads were assembled against the existing *T. pseudonana* genome and all further data processing (e.g. sequence alignment, calling of SNVs) was carried out following the same pipeline as described in [22].

### (g) Statistical analyses

All analyses were carried out in R (v. 3.23–v. 3.51). See the electronic supplementary material for details per analysis.

#### (i) Magnitude of the evolutionary responses

To compare the fold-changes of evolved samples (after 300 generations 't300') compared to the growth rate of the ancestor at the respective selection temperature, the ratios of growth of the ancestor at selection temperature, and the evolved sample at selection temperature were calculated, and a linear mixed model used, where that ratio was the response variable.

#### (ii) Justification for treating subsets of FL-evolved replicates separately

In order to justify our decision to treat the 22°C- and 32°C-preferring replicates from the FL selection environment as separate

groups, we tested whether these replicates would repeatedly cluster with each other when no *a priori* assumption of their identity was made. To do so, phenotypic and molecular (WGS) data were used to construct two Euclidean distance matrices, which served as base for PCoAs constructed within the mixOmics package (6.1–3). For the WGS data, to test for separation of samples by treatment, and within treatment variation, the Euclidean distance matrices were used to perform AMOVAs following [29].

The PCoAs show (electronic supplementary material, figures S9 and S10) that 22°C preferring replicates cluster with the ancestor, and samples evolved under 22°C or 32°C. Replicates preferring the 32°C environment cluster with samples from the FS environment.

#### (iii) Fitness trajectories of FL-evolved lineages

The time series of specific growth rates in FL-evolved lineages were analysed using a generalized additive mixed effects model (GAMM, within the R package gamm4, v. 0.2–5) to assess whether the fitness trajectories differed between the biological replicates within the selection regime depending on the temperature of the selection regime (22°C or 32°C).

#### (iv) Characteristics of thermal tolerance curves of growth

To describe and compare characteristics of the thermal tolerance curves of growth, generalized additive mixed effects models (gamm4 as above) were fitted to the growth rate data across a temperature gradient spanning 15°C to 40°C. Several parameters describe the shapes of the thermal tolerance curves: The peak of the curve, $T_{opt}$ indicates the temperature at which growth is fastest. The breadth of the curve is the range of temperatures at which growth is a given percentage of the peak rate (usually 50–80%). We used these parameters to estimate the area underneath the curve (AUC) and compare it between treatments. The area underneath the thermal tolerance curve is a function of both curve breadth and curve height, and we, therefore, use it here to describe changes in the shape (including height) of thermal tolerance curves among our selection lines.

#### (v) Plasticity of photosynthesis and photochemical traits

Phenotypic plasticity in photosynthesis and photochemical traits was calculated as described above (i.e. *via* comparing the steepness of slopes). The steepness of slope was used as the response variable in a mixed effects model with assay temperature and selection temperature as fixed factors, and biological replicate nested within selection environment as the random factor.

#### (vi) FRRF data

$\Phi_{PSII}$, C and rP, parameters were each extracted from separate nonlinear mixed effects models, where 'selection regime' and 'assay temperature' were fixed effects and replicate was a random effect on the intercept. Model selection proceeded as above and suggested that the best model included selection regime and assay temperature for $\Phi_{PSII}$, C and rP.

## 3. Results and discussion

### (a) Magnitudes and trajectories of evolutionary responses to warming under fast and slow fluctuations

For the scope of this paper, we define an evolutionary response as the growth rate in the selection environment compared to the growth rate or trait value in the ancestor when exposed to the selection environment. On average,

evolutionary responses of growth rates were larger in the fluctuating than in the stable environments (figure 1a; electronic supplementary material, table S1), in line with previous findings [22,30,31]. There was no overall effect of the frequency of fluctuations on the average magnitude of the evolutionary response, but the trajectories by which the fluctuating environments achieved similar magnitudes of responses were strikingly different. Under fast fluctuations (fluctuating short, FS), growth rates quickly increased early on in the selection experiment [22], yielding higher growth rates (measured in the respective selection environment) in samples selected in the fluctuating selection environment than in samples selected at 22°C or 32°C. Growth rates in samples evolving under the slower fluctuations (fluctuating long, FL) were initially up to 1.3-fold lower than growth rates in samples evolving under fast fluctuations. Growth was especially slow during the first 100 generations of evolution (figure 1c; see electronic supplementary material, table S2 for statistics), and exacerbated in the periods of selection at 32°C. Yet, growth rates at 32°C in the FL lines were never reduced to the same level as those under constant selection at 32°C (geometric mean growth rates 0.24 ± 0.1 s.e.m., and 0.31 ± 0.04 s.e.m., for average growth rate of 32°C selected and FL-selected samples at 32°C during the first 100 generations, respectively). After approximately 100 generations, growth rates during periods at 32°C recovered to and above the levels seen in the samples evolving in the 22°C stable environment, and after approximately 150 generations, two distinct populations emerged in the FL treatment (figure 1a,b), with half of the biological replicates consistently growing better at 22°C than at 32°C (geometric mean growth rate at 22°C: 0.61 ± 0.03, geometric mean growth rate at 32°C: 0.57 ± 0.04, ± 1 s.e.m.) and the other half growing better at 32°C than at 22°C (geometric mean growth rate at 32°C and at 22°C: 0.62 ± 0.04 and 0.55 ± 0.04, ± 1 s.e.m respectively). When we use all available phenotype or molecular data to construct principal component analyses with no a-priori assumptions about the behaviour of individual biological replicates, we see that the same replicates cluster together (electronic supplementary material, S9 and S10). The three replicates that favour the 22°C environment cluster closer to the samples evolved under the stable 22°C regime and the three replicates that favour the 32°C environment cluster closer to the samples evolved under rapid fluctuations.

While adaptation to the extreme warm environment in one half of the FL-evolved replicates is in line with our findings under fast fluctuations (see also [22]), adaption to a less extreme temperature, as is the case in the remaining set of FL-evolved replicates, may help to generate high geometric mean fitness in a fluctuating environment [32]. Notably, theoretical frameworks would have predicted adaptive tracking to evolve under the slow and predictable fluctuations of the FL selection environment [19]. When organisms 'track' an environment, their populations keep adapting to the new environmental conditions as they encounter them [19,33]. As evolutionary responses require a minimum number of generations to pass for mutations to arise and fix, the likelihood of organisms tracking the fluctuations of an environment rather than reacting through plasticity increases when the fluctuations span several generations. In our case, the frequencies of the slow fluctuations were longer than generation times (characteristic of a coarse grained environment [34]), but the

amplitude between fluctuations relative to the frequency and the organisms' generation time may have been too large to allow for a solution encompassing largely adaptive tracking. The splitting up of the populations we observe here is more likely in line with diversification bet-hedging, where organisms invest in several different strategies that increase fitness across a range of environments (here, increase temperature tolerance), which is assumed to develop primarily when fluctuations are not as easily predictable [19]. Further, while the evolution of increased plasticity may seem counterintuitive under slow fluctuations, at least one modelling study suggests that in coarse-grained environments, plasticity may evolve as a by-product of inefficient short-term natural selection [35], where, populations evolve long-term adaptive plasticity through the accumulation of limited genetic change during each fluctuation.

## (b) Differences in the frequencies of fluctuations affect evolution of thermal tolerance

To better understand the consequences of these different evolutionary responses within and between fluctuating treatments, we now explore which environments the evolved and ancestral populations can inhabit (their thermal tolerance curves) and the phenotypes they are capable of displaying (their phenotypic plasticity). We found that thermal tolerance curve characteristics differed between stable and fluctuating environments and between environments that fluctuated on short or longer time scales (i.e. 3–4 or 30–40 generations respectively; figure 2a–d; see electronic supplementary material, tables S3–S6): populations evolved in the rapidly fluctuating environments had faster growth at temperatures exceeding 32°C (figure 2a; electronic supplementary material, tables S3 and S6) than samples evolved in stable conditions. Half of the replicates from the environment that fluctuated on longer time scales (FL) behaved similarly to those evolved under rapid fluctuations. The thermal tolerance curves of the remaining FL-evolved samples more closely resembled tolerance curves of samples evolved under stable conditions.

Samples evolved under fluctuations also maintained larger AUCs (figure 2b; likelihood ratio test comparing models with and without 'selection temperature': $\Delta$d.f. = 4, $\chi^2 = 23.01$, $p < 0.001$; see also electronic supplementary material, tables S3 and S6). In other words, while all evolved samples were able to maintain growth across all assay temperatures, samples from the rapidly fluctuating environment grew faster across a larger number of assay temperatures than did samples from the stable environments.

In rapidly fluctuating environments, optimum temperatures (figure 2c; likelihood ratio test comparing models with and without 'selection temperature': $\Delta$d.f. = 5, $\chi^2 = 71.95$, $p < 0.0001$; see also electronic supplementary material, tables S4 and S6), and growth rates at the optimum temperature (figure 2d; likelihood ratio test comparing models with and without 'selection temperature': $\Delta$d.f. = 5, $\chi^2 = 86.11$, $p < 0.0001$; see also electronic supplementary material, tables S5 and S6) were higher than samples evolved at 22° or 32°C under stable conditions. These pronounced changes in thermal tolerance curve characteristics add to the growing body of evidence for the evolvability of tolerance curves in the absence of breadth/height trade-offs (but see [36–38]

**Figure 1.** (*Caption overleaf.*)

**Figure 1.** (*Overleaf.*) Magnitudes of evolutionary responses (*a*) in all selection regimes and (*b*) in the slowly fluctuating selection regime, and (*c*) growth rate trajectories for FL-evolved populations (slow fluctuations). (*a*) Magnitudes of evolutionary responses calculated as the fold difference of growth in the selection environment after 300 generations to growth after less than 20 generations. Values larger than 1 indicate that growth rates after evolution are higher than growth rates elicited through a preliminary plastic response after only a few generations. All boxplots are displayed with the girdle indicating the median, and whiskers extending to the 25th and 75th percentile. *n* = 6 per selection environment throughout. (*b*) The subpanel displays the two distinct populations from the FL regime. All boxplots are displayed with the girdle indicating the median, and whiskers extending to the 25th and 75th percentile. *n* = 6 per selection environment throughout. (*c*) Growth rate trajectories displayed for 400 generations of selection in the FL environment. Between 100 and 200 generations, average growth rate increases, and there are two distinct populations. One grows faster at 32℃ than at 22℃, while the other grows faster at 22℃ than at 32℃. Trajectories were fitted with a GAMM. Triangles are for the three replicates that preferred the 32℃ environment, circles, for the three replicates that preferred the 22℃ environment. The shade of the symbols indicates the selection environment temperature (lighter for 32℃ and darker for 22℃). (Online version in colour.)

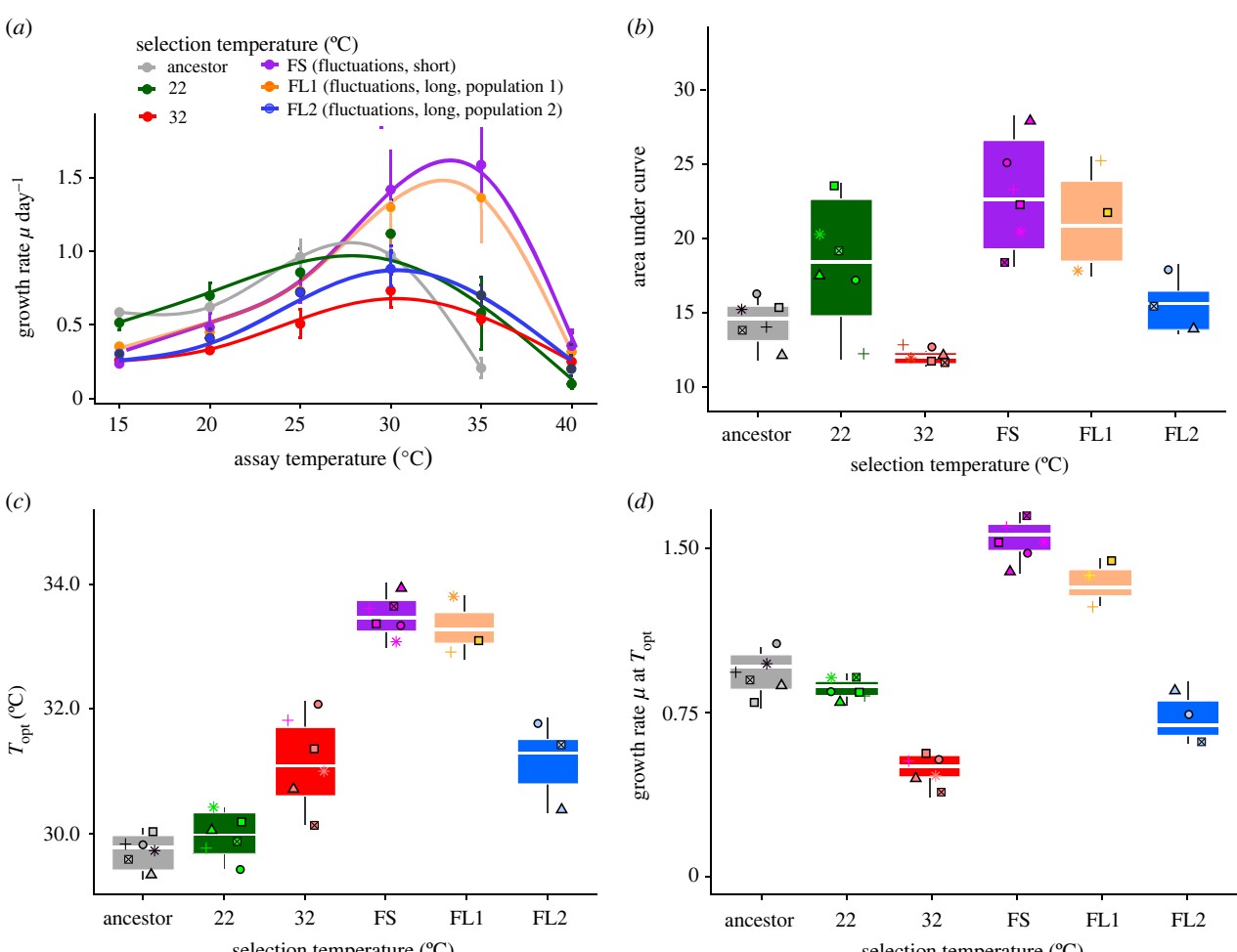

**Figure 2.** Characteristics of thermal tolerance curves evolving in environments cycling between 22℃ and 32℃ rapidly (3–4 generations, purple) or slowly (approx. 30–40 generations, orange for 32℃-preferring population, and blue for the 22℃ preferring population), in the stable environments framing the fluctuations (22℃ and 32℃), and in the ancestor. (*a*) Growth rates (μ day$^{-1}$) in the ancestor and all evolved lineages across a temperature gradient spanning 15℃ to 40℃. Displayed are means ± 1 s.e.m. Curves were fitted via a GAMM model with three inflection points. (*b*) Area under the curves fitted in (*a*). The area underneath the curve is larger under rapid than under slow fluctuations, and in either case larger than in the stable environments framing the fluctuations. (*c*) $T_{opt}$ (℃) as the temperature where growth rate is highest, extracted from the curves in (*a*). Samples in fluctuating environments evolve the highest $T_{opt}$ values, with samples from the rapidly fluctuating environment displaying significantly higher $T_{opt}$ than samples from the slowly fluctuating environment. (*d*) Growth rates at $T_{opt}$ as determined for panel (*c*). All boxplots are displayed with the girdle indicating the median, and whiskers extending to the 25th and 75th percentile. *n* = 6 per selection environment throughout. (Online version in colour.)

for examples where trade-offs do underpin changes in the curve characteristics).

Samples evolved under slower fluctuations again showed two distinct patterns on the basis of their thermal tolerance characteristics (figure 2*a–d*). Half of the biological replicates (growing better at 32℃ than 22℃) showed curve

characteristics similar to those found in samples evolved under fast fluctuations, with higher AUCs, higher $T_{opt}$, and higher peak rates at $T_{opt}$ than samples evolved in stable environments. The other half of the biological replicates (growing better at 22℃ than 32℃) evolved AUCs and rates at optimum temperatures more akin to those in the 22℃

environment, however, $T_{opt}$ shifted to warmer temperatures markedly, such that optimum temperatures were similar to those found in 32°C-evolved samples.

## (c) Evolution of phenotypic plasticity under rapid and slow fluctuations

To link thermal tolerance to trait plasticity, we need to first establish a measure of plasticity that can be used across different phenotypic traits. Phenotypic plasticity is well-defined for linear reactions norms [11], where the steepness of the slope indicates the magnitude of plasticity [39–41]. Here, we measured labile metabolic traits (i.e. traits that change reversibly within the lifetime of an organism [21]) across a thermal gradient in the ancestral and evolved samples across a range of temperatures to (i) test whether plasticity in metabolic traits evolves and (ii) whether the degree to which it does evolve hinges on the frequencies of fluctuations. For photosynthesis and respiration, we consider the plastic response of metabolic traits to be within temperatures spanning the fluctuations (i.e. from 22°C to 32°C), which is the linear part of the unimodal growth curve for ln-transformed rate values. We then treat the steepness of the slopes across this gradient as a measure of plasticity. We report data at the population level as it is impossible to measure these traits on the level of a single cells with current technology. However, aliquots of all samples were made clonal again at the end of the experiment, and growth rate measurements suggest very little within-replicate variation (electronic supplementary material, figure S6).

The selection environment determined the steepness of the slopes of photosynthesis (figure 3a, likelihood ratio test comparing models with and without 'selection temperature': $\Delta$d.f. = 5, $\chi^2$ = 21.53, $p < 0.001$ see also electronic supplementary material, table S7 and electronic supplementary material, figure S2). The steepest slopes (i.e. the largest plastic response) evolved in samples selected in the rapidly fluctuating environment, and the warm-preferring subset of samples selected in the slowly fluctuating environment. To determine whether the evolution of high plasticity in the fast fluctuations and the presence of distinct phenotypes under slow fluctuations were reflected on a photochemical level, we used fast repetition rate fluorometry [25] (FRRF; figure 3c,d; electronic supplementary material, figures S3–S5), and assayed the evolved samples at 22°C and 32°C. We determined trait plasticity as described above for the following parameters: (i) $F_V/F_M$ as the maximum quantum efficiency of photosystem II (figure 3b; electronic supplementary material, figure S3); (ii) rP at $I_{opt}$, which describes the relative rate of photosynthesis as the amount of electron transport through photosystem II at the optimum light intensity (figure 3c; electronic supplementary material, figure S4); and (iii) $\Phi$PSII, as a measure of the operating photochemical efficiency of PSII (figure 3d; electronic supplementary material, figure S5). We find, again, that samples from the rapidly fluctuating FS environment and the 32°C-preferring subset of the samples evolved under the slow fluctuations displayed higher plasticity than samples from the stable environments (electronic supplementary material, tables S7–S10). Although there was some variation in plasticity in the ancestor, ancestral plasticity is not indicative of the amount to which plasticity evolves across the different treatments in our case.

Theory [19] and some empirical studies [30] predict that enhanced plasticity should evolve under rapid fluctuations relative to the organisms' generation time. Here, we show that fluctuations need not be rapid for the evolution of elevated phenotypic plasticity. Phenotypic plasticity evolved in half of our samples evolved under fluctuations on longer time scales (i.e. plastic and fixed strategies seem to be able to evolve with similar frequencies on the time scales chosen for this experiment). The evolution and maintenance of alternative phenotypes has a rich theoretical backdrop [12,28,42], though largely with a focus on longer lived organisms with distinct developmental stages, where the maintenance of differing phenotypes hinges on environmental variability occurring at a certain point in an organism's development (usually early development). While we cannot say for sure why two different strategies evolved here, we can suggest that they lead to approximately equal fitness, with chance mutational events determining a specific outcome. Evolution of higher plasticity under the slow fluctuations may be a correlated response of selection on changing a mean trait value to minimize fitness variance as in conservative bet-hedging or a mix of strategies to take advantage of alternative environmental scenarios in a more probabilistic fashion (diversification bet-hedging) [43].

There were marked differences in the degree to which plasticity evolved even under fast fluctuations, ranging from a 1.5 to a 2.9 fold increase in plasticity compared to the ancestor. While there are physiological constraints or limits to how plastic an organism can be, the amount to which plasticity evolves can also hinge on whether evolving or maintaining a plastic phenotype carries a cost [44–46]. To estimate whether evolved plasticity is costly in this setting, we carried out reciprocal transplant assays. Samples from the fluctuating environments were assayed under stable conditions at the temperature at either end of the fluctuation (i.e. at 22°C and at 32°C), and likewise, samples evolved under stable conditions were assayed in the fluctuating (short) environment and under all stable conditions. If plasticity itself carried a direct cost, we would expect the more phenotypically plastic samples from the fluctuating selection regime to fare worse under stable conditions than those with lower phenotypic plasticity, and also fare worse than samples selected and assayed under stable conditions. Samples with high plasticity and fast growth rates in their selection environment do not show a marked reduction in growth when assayed in a reciprocal or ancestral environment (electronic supplementary material, figures S6 and S7, electronic supplementary material, table S11). This result points to the cost of plasticity being immeasurable in this setting, in line with previous studies [21,46]. Trade-offs in our setting might either be truly absent, or affect a trait not investigated here (e.g. sexual rather than clonal reproduction), or only become visible in environments not tested here (e.g. truly detrimental environments). Nevertheless, our findings of costs so small that they are immeasurable indicates that the costs of plasticity, when included in models [17,47], may be easily overestimated relative to measurable costs.

## (d) Phenotypic plasticity and thermal tolerance are linked in fluctuating environments

Developing a mechanistic understanding of the interactions between trait plasticity, fitness, and the environment is key to predicting which taxa and phenotypes will thrive in a warming world and how they will contribute to ecosystem functioning.

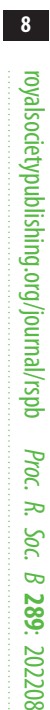

**Figure 3.** Phenotypic plasticity for photosynthesis measured as (*a*) $O_2$ evolution at PSII, (*b*) $F_V/F_M$ as maximum quantum efficiency of PSII after dark incubation, (*c*) rP (rates of electron transport through PSII at $I_{opt}$) and (*d*) $\Phi$ PSII (photochemical efficiency of PSII) in ancestral and evolved lineages. For all traits, plasticity evolves and is on average higher in the fluctuating environments than in the stable environments, and within the fluctuating treatments, the highest phenotypic plasticity evolves under rapid fluctuations, and in the 32°C preferring subpopulation of samples from the slow fluctuating treatment. Negative values in the ancestor or evolved samples owe to ancestor's having a thermal optimum at temperatures lower than 32°C. Shapes are for the biological replicates. Boxplots are displayed with the girdle indicating the median, and whiskers extending to the 25th and 75th percentile. *n* = 6. (Online version in colour.)

The area underneath the thermal tolerance curve (AUC) describes changes in the shape of the thermal tolerance curve, and the AUC value can be used as a composite measure of the fitness–environment interaction. We found that in populations from fluctuating environments, trait plasticity increases linearly as a function of the AUC (figure 4*a*–*d* for linking AUC to trait plasticity in gross photosynthesis (*a*), $F_V F_M$ (*b*), rP at $I_{opt}$ (*c*) and $\Phi$PSII (*d*)). Populations evolved under rapid fluctuations generally have the highest AUC, and highest plastic responses, whereas samples evolved under stable conditions cluster with the ancestor at lower AUC values and lower levels of plasticity. The populations affiliated with 32°C, evolved under slow fluctuations cluster with those from the rapid fluctuation treatment, while the populations affiliated

with 22°C cluster with populations evolved under stable conditions at 22°C and 32°C (see also electronic supplementary material, table S11). The relationship between AUC and plasticity is less pronounced in samples evolved in the stable environments and the subset of samples from the FL selection environment that did not evolve increased plasticity.

## 4. Conclusion

To make predictions about the ecological function of marine microbial primary producers in a changing world, we need to know how the frequencies of environmental fluctuations affect the current and projected scope of phenotypic plasticity and the degree to which phenotypic plasticity carries a

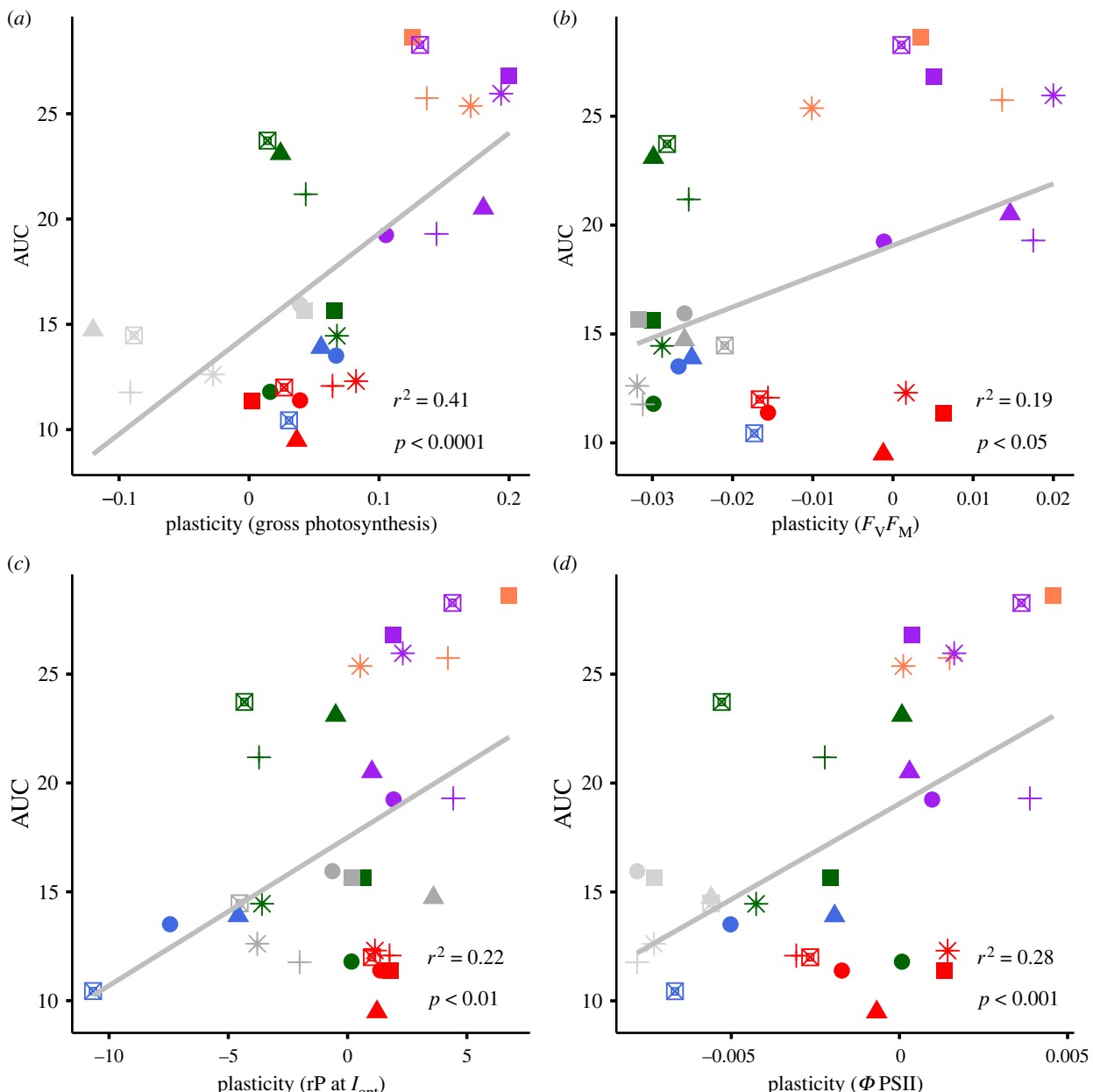

**Figure 4.** Area underneath the tolerance curve for ancestral and evolved samples as a function of trait plasticity in photosynthesis measured as (*a*) $O_2$ evolution at PSII, (*b*) $F_V/F_M$ as maximum quantum efficiency of PSII after dark incubation, (*c*) rP (rates of electron transport through PSII at $I_{opt}$) and (*d*) $\Phi$ PSII (photochemical efficiency of PSII) as a function of. Trait plasticity and tolerance (as area underneath the curve) are positively correlated, such that samples that evolved to have higher trait plasticity also have higher tolerance, which is predominantly the case in samples from the FS environment, and 32°C-preferring samples from the FL environment. The correlation is the most pronounced for plasticity in photosynthesis rates measured as $O_2$ evolution at PSII and in the plastic responses for the photochemical efficiency of PSII. The fitted line is the output of a linear mixed effect model. Shapes indicate individual biological replicates, with $n = 6$ per treatment. (Online version in colour.)

fitness benefit under different frequencies of fluctuation. While there is some experimental evidence indicating that fast-lived phytoplankton from regions that are more variable tend to exhibit larger plastic responses [48,49], no study to date has explicitly linked the effects of evolving under rapid or slow fluctuations in temperature to the evolution of thermal tolerance (growth) and thermal performance (trait plasticity). These aspects, however, are crucial if we are to make predictions about how conditions of today modulate phytoplankton responses in the future, where the average temperature and variation around the mean temperature will increase, so that environments that are now rare may become more common [47,50]. Our results suggest that, on a population level, (i) plasticity evolves easily under rapid, predictably fluctuations, (ii) various stable strategies,

including enhanced plasticity can evolve under slower, predictable fluctuations, and (iii) regardless of fluctuation frequency, enhanced trait plasticity is directly linked to higher fitness, so that populations that experience fluctuating thermal regimes today may be better equipped to deal with further changes to the environment via enhanced plasticity *and* tolerance. Rapid fluctuations in particular might be providing a stepping stone to fast growth and metabolic function at more extreme temperatures that exceed the mean of variation of the environments that they inhabit today.

Data accessibility. Data necessary to re-create the figures and run all statistical tests can be found on Zenodo: https://doi.org/10.5281/zenodo.6394982 [51].

Electronic supplementary material is available online [52].

**Authors' contributions.** C.-E.S.: conceptualization, data curation, formal analysis, investigation, methodology, writing—original draft, writing—review and editing; A.B.: funding acquisition, writing—review and editing; N.S.: funding acquisition, writing—review and editing; G.Y.-D.: conceptualization, funding acquisition, writing—original draft, writing—review and editing.

All authors gave final approval for publication and agreed to be held accountable for the work performed therein.

**Conflict of interest declaration.** We declare we have no competing interests.

**Funding.** This study was funded by a Leverhulme Trust research grant (RPG-2013-335). Exeter Squencing Services are supported by Medical Research Council Clinical Infrastructure award (MR/M008924/1), Wellcome Trust Institutional Strategic Support Fund (WT097835MF), Wellcome Trust Multi User Equipment Award (WT101650MA) and BBSRC LOLA award (BB/K003240/1).

**Acknowledgements.** Whole genome re-sequencing was carried out at Exeter Sequencing Service and Computational core facilities at the University of Exeter, where Dr Karen Moore, Dr Audrey Farbos, Paul O'Neill and Dr Konrad Paszkiewicz lead the handling of the samples. Prof Dr David Studholme advised on WGS analyses and wrote the pipelines.

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
