## [Peer Review File · Proceedings of the Royal Society B: Biological Sciences]

Review History

RSPB-2021-1418.R0 (Original submission)

Review form: Reviewer 1

Recommendation

Reject – article is scientifically unsound

Scientific importance: Is the manuscript an original and important contribution to its field?

Marginal

General interest: Is the paper of sufficient general interest?

Acceptable

Quality of the paper: Is the overall quality of the paper suitable?

Poor

Is the length of the paper justified?

No

Should the paper be seen by a specialist statistical reviewer?

No

Do you have any concerns about statistical analyses in this paper? If so, please specify them explicitly in your report.

No

It is a condition of publication that authors make their supporting data, code and materials available - either as supplementary material or hosted in an external repository. Please rate, if applicable, the supporting data on the following criteria.

Is it accessible?

N/A

Is it clear?

N/A

Is it adequate?

N/A

Do you have any ethical concerns with this paper?

Yes

Comments to the Author

This manuscript reports the result of a 300-generation experimental evolution study designed to understand how stable and fluctuating temperature environments influence the evolution of plasticity and tolerance curves in the model diatom *Thalassiosira pseudonana*. The main results reported are effects of treatment on the magnitude of the evolutionary response, a split in the response of the replicates in the 30-40 generation fluctuation treatment, an effect of environment on thermal tolerance curves, an effect of treatment on the degree of plasticity shown by photochemical parameters, details about the patterns of molecular evolution demonstrated in the different treatments. Overall, this seems like a worthy and interesting study but I struggled to follow the hypothesis being tested and how it relates to theory that is frequently alluded to and I have some major concerns about the novelty of the work.

1. The study is presented as if this is a new study but there seems to be a lot of overlap with the authors previous 2018 paper (Nature Communications, 1–14. (doi:10.1038/s41467-018-03906-5) which already demonstrated an increase in growth rates at or near selection temperatures, an effect of fluctuating selection on evolutionary response, phenotypic change in metabolic traits and elemental composition, rapid evolutionary shifts in thermal tolerance curves and the molecular divergence between treatments that underpin these differences. It seems that these are the same results presented differently.
2. This manuscript reports the results from a slow fluctuation treatment (30-40 generations) that weren't reported in the previous study. A big focus of this study is supposedly comparing what effect 30-40 generation fluctuation has compared to the 3-4 generation effects (previously reported in the 2018 paper). The paper talks about long and short fluctuations leading to different predictions for plasticity, tolerance curves and AUC's but I was a bit confused as to how these predictions were reached and how they would then be directly tested. The results don't stick closely to testing these predictions and much of the paper refers to a split in the replicates from the long-fluctuations treatment but no sensible rationale for analysing the data as a split result is given (although it is clear that is what has happened).
3. The manuscript suggest that theory explains the link between plasticity and evolutionary potential but then doesn't explain what that link is clearly or how we should expect stable and fluctuating environments of different duration (relative to generation time) to influence it. To look at the evolution of plasticity you would need to isolate single clones and measure plastic responses in the ancestor (single clone) and then compare those responses with the plastic responses of clonal isolates from the evolved populations. But here the authors are just assessing population level plasticity. In this case, it's not possible to know whether the differences observed derive from plasticity or changes in the frequency of clones in the population in the different environments.

4. I couldn't follow how you generate estimate of plasticity in photosynthesis. A figure might help to explain how you get a measure of photosynthesis in each environment from fitting NP data to a dynamic model of photoinhibition via non-linear least squares regression.

5. There isn't ever really any attempt to justify the treatments used in relation to natural environments. Given that the experiment started with a single clone from a lab culture it will presumably have adapted to that static 22oC environment. The possible consequences of this should be discussed in the paper. Is not possible to isolate clones from the wild?

Minor comments

Ln 39-41 – logic not clear to me

Ln 49-50 – not sure what you mean here? How is evolutionary flexibility got anything to do with being a good descriptor for thermal tolerance?

Ln 56-59 – this is all too vague and doesn't explain to the reader what hypothesis needs testing.

Ln 67-72 – these are quite clear predictions but not clearly justified. The results section doesn't stick very closely to what is put forward here.

Ln 86 – you don't give a reference to your previous study here, you need to.

Ln 308-309 – adaptive plasticity isn't expected to evolve in unpredictable environments.

Ln 323 – is there a reference to support your statement?

Ln 344-345 – Given that you haven't previously explained the difference between a cost of plasticity and a limit of plasticity this sentence may confuse readers.

Ln349 – 351 – Not clear

Fig. 1B – is this the wrong figure?

Fig. 4 – these correlations don't always appear to hold within treatments. Is it really appropriate to propose a general relationship that is largely a result of treatment effects?

Review form: Reviewer 2

Recommendation

Major revision is needed (please make suggestions in comments)

Scientific importance: Is the manuscript an original and important contribution to its field?

Good

General interest: Is the paper of sufficient general interest?

Good

Quality of the paper: Is the overall quality of the paper suitable?

Poor

Is the length of the paper justified?

Yes

Should the paper be seen by a specialist statistical reviewer?

No

Do you have any concerns about statistical analyses in this paper? If so, please specify them explicitly in your report.

Yes

It is a condition of publication that authors make their supporting data, code and materials available - either as supplementary material or hosted in an external repository. Please rate, if applicable, the supporting data on the following criteria.

Is it accessible?

Yes

Is it clear?

No

Is it adequate?

No

Do you have any ethical concerns with this paper?

No

Comments to the Author

In the time I had available for this review I was not able to work through the entire manuscript and supplementary information. I apologise that I could not devote more time to the task.

The submitted material concerns what appears to be a well-motivated experimental study of evolution of thermal performance curves in constant and fluctuating temperature environments. The subject matter is, in my opinion, of general interest to ecologists and evolutionary biologists, and is therefore likely of interest to the general audience of PRSB.

Nevertheless, there are many things that can and need to be addressed. The standard of presentation of the research appears to be insufficient for publication. Important elements of the experimental design are not given. Quantitative methods are unclear, and some seem inappropriate. Not all the material presented seems necessary given the Introduction. Some results stated in the text seem to be contradicted by results presented in the figures and analysis tables.

I give below details of some of these issues, and of other issues, but please recall that I could not finish working through the entire ms in the time I had available. Therefore please extrapolate my findings to the conclusion that all aspects of the submitted material should be thoroughly and critically reviewed and revised before any resubmission.

Point-by-point comments follow:

Abstract line 19: please consider stating how the frequency determines the shape of the tolerance curve, rather than just that it does.

Line 38. It is unclear what the "this" refers to, since the contents of the previous sentence do not clearly create the implication of constant area. And please consider referencing evidence for the statement.

Line 42. Please be more specific about what are "these assumptions". Currently a reader has to try to sort them out from the previous text, which is not ideal.

Line 86, please consider referencing evidence for the statement.

The genomic investigation appears to have little or no conceptual basis presented in the Introduction. Same goes for the photochemical traits work. This is a theme that recurs: e.g. tracking and analysing fitness trajectories is not well justified in the Introduction. More generally, it seems that the submitted manuscript is more of a report of everything that was done in project, compared to a reporting of only the results required to address the matters raised in the Introduction. A critical review of content seems appropriate.

The nature of the "cycling" between 22 and 32 is unclear. Was it a sinusoidal cycle? Or was it switching between the two temperatures? This seems important, as a square-wave type

fluctuation seems much less relevant than a smooth one. Also, how were treatments and replicates distributed among incubators (particularly the three FL1 and three FL2 replicates)?

Line 182 states “Phenotypic plasticity was calculated as described above...”. I’m sorry if I missed something obvious, but I don’t see where this was described “above”.

There is considerable duplication of text between some sections of the main text and the supplementary information. It then takes more time than it should to find any unique information (e.g. about the nature of the fluctuations).

Line 171. Why not ln-normalised?

Figure 1A and inset. Please display all data points. And there is little point in having the different colours; it might even be confusing to have them. And on the version of the figure with the caption below there is no numeric scale or dashed line at $y=1$ on the inset. And please help the reader translate between generations and days. E.g. put number of generations on the x-axis, rather than number of days.

Analysis of the evolutionary response (Table 1). The sentence starting “We fitted a...” implies that growth rate of the evolved sample was the response variable. The next sentence, however, suggests the response variable might have been a relative growth rate. Thus I am already confused. The description goes on to state that a full model with selection environment as a fixed factor and biological replicate as a random factor was used. This implies that there were repeated measures, i.e. multiple measures for each biological replicate. But this doesn’t correspond with the context of Figure 1A, where it is only the terminal response. Finally, why was model selection performed at all? There is only one fixed factor (treatment)... why not just use an F-test (or likelihood ratio test) on the treatment. Why was model selection used in each of the analyses it was used in?

Line 238-244 content would be better placed in the Methods, or even Introduction.

Line 247 states “Populations evolved in the fluctuating environments had faster growth at temperatures exceeding 32°C (Fig 2 A, Supporting Tables 3 and 6) than samples evolved in stable conditions.” But Figure 2A and 2D shows that the FL2 replicates did not have higher growth rate above 32°C than samples evolved in stable conditions. That is, the result stated in the text is not supported by the data presented in the figures. This discrepancy also appears to occur in the following two paragraphs, one about AUC and one about optimum temperature.

Finally, I took a quick look at the zenodo code and data submission, and tried to run the 20210610_Thally02_Figure02_plot_and_stats.R script.... Line 35 is commented out, but would read a required dataset. I could not find on zenodo the required file, so could not run line 35 in any case. It seems, therefore, that the submitted code cannot be run, at least for this analysis. And a suggestion: instead of writing “Use at your own risk” write something like “Although the code is guaranteed to appropriately and accurately analyse our experimental data to produce the results presented in the published report, any reuse or adaptation of this code for other situations is at own risk.” and put this in each of the code files.

Review form: Reviewer 3

Recommendation

Accept with minor revision (please list in comments)

Scientific importance: Is the manuscript an original and important contribution to its field?

Excellent

General interest: Is the paper of sufficient general interest?

Excellent

Quality of the paper: Is the overall quality of the paper suitable?

Excellent

Is the length of the paper justified?

Yes

Should the paper be seen by a specialist statistical reviewer?

No

Do you have any concerns about statistical analyses in this paper? If so, please specify them explicitly in your report.

No

It is a condition of publication that authors make their supporting data, code and materials available - either as supplementary material or hosted in an external repository. Please rate, if applicable, the supporting data on the following criteria.

Is it accessible?

Yes

Is it clear?

Yes

Is it adequate?

Yes

Do you have any ethical concerns with this paper?

No

Comments to the Author

This is a wonderful piece of work addressing a set of compelling questions that tie together theory about evolution and plasticity with understanding the pace and direction of trait responses to climate change in a group of organisms central to ecosystem function.

There are a few minor issues to address.

Line 67-72. Careful with terminology here.

I think genetic adaptation (e.g. via allele frequency change) is normally associated with polymorphism - the existence of different genetically based phenotypes. Plasticity, where the genotypes possess alternative phenotypes, is, I think, polyphenism. It feels like this section doesn't distinguish the mechanism and processes very well and mixes and matches the concepts and terms.

Line 93-94 - this is the first instance of a wider issue about the inclusion of the whole genome re-sequencing data. These data lack the theoretical/conceptual motivation of the rest of the work - even here, the motivation is to link the classy experimental findings with genomic data. This isn't a motivation reflected by the rest of the work where theory underpins expected pattern in specific types of data. As such the genomic data presented don't test any hypotheses and are disconnected from the main questions about evolution and plasticity. It either needs much more development and predictions about what might be expected (e.g. single genes/many genes,

classes of genes that might be enriched, links to plasticity vs. allele frequency change) or removal.

Line 140 (Plasticity) - This describes the measurements made, but not the reaction norm. It's clear that the growth rate thermal performance curves are reaction norms and plasticity. What are the 'genotypes' here and the environmental gradient? Needs some additional clarification about where the slope used to define plasticity comes from and motivation. Comparing ancestor to evolved (which this starts with) doesn't align with classic definitions of plasticity arising from variable environments.

Line 150 - This is rather sparse description of the genomic data pipeline. It's also not clear what the endpoints are, why they are being estimated and how key questions are answered (see above)

Line 181 (Plasticity) - lacks details to understand which 'above' method the MS is referring to.

Line 191 - why are the genomic data stats with that section and not in the stats section?

Line 200 - Is this pattern also true in non-microbes? Any evidence to generalise the patterns a bit more?

Line 217 - This feels like one of the most significant results. Would the MS benefit from a more precise definition of population here? Is the phenotypic description? Is this result worth isolating so that the investigation of plasticity vs polymorphism stands out a bit more?

Line 229 - 234 - The bet-hedging theory is a good template. Is it also worth discussion theory about polymorphisms relative to the grain of the environment (the classic literature).

Distinguishing how the pattern is more or less associated with each mechanism associated with how polymorphisms can arise would be valuable. Perhaps introducing more clearly the idea of adaptive tracking, how it is affected by the pace and grain of environment variation and then how the data in the MS on allele frequencies and plasticity fit in might be good.

Line 264 - It would be nice to have a bit more insight in to the arguments about why these expected trade-offs might be relaxed, whether the trade-offs exist in a different biological space, and what the conundrum is if there is no detectable trade-off.

Line 278-280 - This description would be valuable in the methods.

Line 291 - 306 - There are a lot of methods here. Not clear why these are not in the methods..... (see above)

Line 311 - This is a really compelling and interesting result. There is a lot of old theory - Moran 1992 - The evolutionary maintenance of alternative phenotypes; Lively's work; West Eberhard - that might be worth revisiting here?

Line 336 - Is it worth separating out the issue of costs? It's been such an industry that these results deserve a bit of a focus? Does this result also tie into to the lack of trade-offs?

Line 347 - As above - these data and results 'dangle' from the rest of the MS because a firm connection to theory is missing. Can these data be tied more closely to the phenotypic work above and theory so that patterns of relatedness based on SNPS, and the enrichment exercise are more informative? One might also have expected some insight from allele frequency data?

One solution is just to leave these data out. If not, more substantial motivation and theory is needed.

Decision letter (RSPB-2021-1418.R0)

02-Aug-2021

Dear Dr Schaum:

I am writing to inform you that your manuscript RSPB-2021-1418 entitled "Rapid evolution of thermal tolerance and phenotypic plasticity in variable environments" has, in its current form, been rejected for publication in Proceedings B.

This action has been taken on the advice of referees, who have recommended that substantial revisions are necessary. With this in mind we would be happy to consider a resubmission, provided the comments of the referees are fully addressed. However please note that this is not a provisional acceptance.

Sincerely,
Professor Hans Heesterbeek
<mailto:proceedingsb@royalsociety.org>

Associate Editor
Board Member: 1
Comments to Author:

Your ms has been assessed by 3 expert reviewers, each of whom has provided a detailed and thoughtful report. All of the reviewers saw value in the experiments. However, although the reviewers' vary in their overall assessments of the suitability of the work for PRSB, they each raise substantive critiques of the work as it is presented. There appear to be some important conceptual issues that require more attention (e.g. whether you are really measuring plasticity of genotypes?). In addition, 2 of the reviewers questioned whether there was sufficient rationale for the genomic work. There are issues throughout in terms of the clarity of the writing and presentation, particularly with respect to quantitative and bioinformatic methods.

Reviewer(s)' Comments to Author:

Referee: 1

Comments to the Author(s)

This manuscript reports the result of a 300-generation experimental evolution study designed to understand how stable and fluctuating temperature environments influence the evolution of plasticity and tolerance curves in the model diatom *Thalassiosira pseudonana*. The main results reported are effects of treatment on the magnitude of the evolutionary response, a split in the response of the replicates in the 30-40 generation fluctuation treatment, an effect of environment on thermal tolerance curves, an effect of treatment on the degree of plasticity shown by photochemical parameters, details about the patterns of molecular evolution demonstrated in the different treatments. Overall, this seems like a worthy and interesting study but I struggled to follow the hypothesis being tested and how it relates to theory that is frequently alluded to and I have some major concerns about the novelty of the work.

1. The study is presented as if this is a new study but there seems to be a lot of overlap with the authors previous 2018 paper (Nature Communications, 1-14. (doi:10.1038/s41467-018-03906-5) which already demonstrated an increase in growth rates at or near selection temperatures, an effect of fluctuating selection on evolutionary response, phenotypic change in metabolic traits and elemental composition, rapid evolutionary shifts in thermal tolerance curves and the molecular divergence between treatments that underpin these differences. It seems that these are the same results presented differently.
2. This manuscript reports the results from a slow fluctuation treatment (30-40 generations) that weren't reported in the previous study. A big focus of this study is supposedly comparing what effect 30-40 generation fluctuation has compared to the 3-4 generation effects (previously reported in the 2018 paper). The paper talks about long and short fluctuations leading to different predictions for plasticity, tolerance curves and AUC's but I was a bit confused as to how these predictions were reached and how they would then be directly tested. The results don't stick closely to testing these predictions and much of the paper refers to a split in the replicates from the long-fluctuations treatment but no sensible rationale for analysing the data as a split result is given (although it is clear that is what has happened).
3. The manuscript suggest that theory explains the link between plasticity and evolutionary potential but then doesn't explain what that link is clearly or how we should expect stable and fluctuating environments of different duration (relative to generation time) to influence it. To look at the evolution of plasticity you would need to isolate single clones and measure plastic responses in the ancestor (single clone) and then compare those responses with the plastic responses of clonal isolates from the evolved populations. But here the authors are just assessing population level plasticity. In this case, it's not possible to know whether the differences observed derive from plasticity or changes in the frequency of clones in the population in the different environments.
4. I couldn't follow how you generate estimate of plasticity in photosynthesis. A figure might help to explain how you get a measure of photosynthesis in each environment from fitting NP data to a dynamic model of photoinhibition via non-linear least squares regression.
5. There isn't ever really any attempt to justify the treatments used in relation to natural environments. Given that the experiment started with a single clone from a lab culture it will presumably have adapted to that static 22oC environment. The possible consequences of this should be discussed in the paper. Is not possible to isolate clones from the wild?

Minor comments

Ln 39-41 – logic not clear to me

Ln 49-50 – not sure what you mean here? How is evolutionary flexibility got anything to do with being a good descriptor for thermal tolerance?

Ln 56-59 – this is all too vague and doesn't explain to the reader what hypothesis needs testing.

Ln 67-72 – these are quite clear predictions but not clearly justified. The results section doesn't stick very closely to what is put forward here.

Ln 86 – you don't give a reference to your previous study here, you need to.

Ln 308-309 – adaptive plasticity isn't expected to evolve in unpredictable environments.

Ln 323 – is there a reference to support your statement?

Ln 344-345 – Given that you haven't previously explained the difference between a cost of plasticity and a limit of plasticity this sentence may confuse readers.

Ln349 – 351 – Not clear

Fig. 1B – is this the wrong figure?

Fig. 4 – these correlations don't always appear to hold within treatments. Is it really appropriate to propose a general relationship that is largely a result of treatment effects?

Referee: 2

Comments to the Author(s)

In the time I had available for this review I was not able to work through the entire manuscript and supplementary information. I apologise that I could not devote more time to the task.

The submitted material concerns what appears to be a well-motivated experimental study of evolution of thermal performance curves in constant and fluctuating temperature environments. The subject matter is, in my opinion, of general interest to ecologists and evolutionary biologists, and is therefore likely of interest to the general audience of PRSB.

Nevertheless, there are many things that can and need to be addressed. The standard of presentation of the research appears to be insufficient for publication. Important elements of the experimental design are not given. Quantitative methods are unclear, and some seem inappropriate. Not all the material presented seems necessary given the Introduction. Some results stated in the text seem to be contradicted by results presented in the figures and analysis tables.

I give below details of some of these issues, and of other issues, but please recall that I could not finish working through the entire ms in the time I had available. Therefore please extrapolate my findings to the conclusion that all aspects of the submitted material should be thoroughly and critically reviewed and revised before any resubmission.

Point-by-point comments follow:

Abstract line 19: please consider stating how the frequency determines the shape of the tolerance curve, rather than just that it does.

Line 38. It is unclear what the "this" refers to, since the contents of the previous sentence do not clearly create the implication of constant area. And please consider referencing evidence for the statement.

Line 42. Please be more specific about what are "these assumptions". Currently a reader has to try to sort them out from the previous text, which is not ideal.

Line 86, please consider referencing evidence for the statement.

The genomic investigation appears to have little or no conceptual basis presented in the Introduction. Same goes for the photochemical traits work. This is a theme that recurs: e.g. tracking and analysing fitness trajectories is not well justified in the Introduction. More generally, it seems that the submitted manuscript is more of a report of everything that was done in project, compared to a reporting of only the results required to address the matters raised in the Introduction. A critical review of content seems appropriate.

The nature of the "cycling" between 22 and 32 is unclear. Was it a sinusoidal cycle? Or was it switching between the two temperatures? This seems important, as a square-wave type fluctuation seems much less relevant than a smooth one. Also, how were treatments and replicates distributed among incubators (particularly the three FL1 and three FL2 replicates)?

Line 182 states “Phenotypic plasticity was calculated as described above...”. I’m sorry if I missed something obvious, but I don’t see where this was described "above".

There is considerable duplication of text between some sections of the main text and the supplementary information. It then takes more time than it should to find any unique information (e.g. about the nature of the fluctuations).

Line 171. Why not ln-normalised?

Figure 1A and inset. Please display all data points. And there is little point in having the different colours; it might even be confusing to have them. And on the version of the figure with the caption below there is no numeric scale or dashed line at $y=1$ on the inset. And please help the reader translate between generations and days. E.g. put number of generations on the x-axis, rather than number of days.

Analysis of the evolutionary response (Table 1). The sentence starting “We fitted a...” implies that growth rate of the evolved sample was the response variable. The next sentence, however, suggests the response variable might have been a relative growth rate. Thus I am already confused. The description goes on to state that a full model with selection environment as a fixed factor and biological replicate as a random factor was used. This implies that there were repeated measures, i.e. multiple measures for each biological replicate. But this doesn’t correspond with the context of Figure 1A, where it is only the terminal response. Finally, why was model selection performed at all? There is only one fixed factor (treatment)... why not just use an F-test (or likelihood ratio test) on the treatment. Why was model selection used in each of the analyses it was used in?

Line 238-244 content would be better placed in the Methods, or even Introduction.

Line 247 states “Populations evolved in the fluctuating environments had faster growth at temperatures exceeding 32°C (Fig 2 A, Supporting Tables 3 and 6) than samples evolved in stable conditions.” But Figure 2A and 2D shows that the FL2 replicates did not have higher growth rate above 32°C than samples evolved in stable conditions. That is, the result stated in the text is not supported by the data presented in the figures. This discrepancy also appears to occur in the following two paragraphs, one about AUC and one about optimum temperature.

Finally, I took a quick look at the zenodo code and data submission, and tried to run the 20210610_Thally02_Figure02_plot_and_stats.R script.... Line 35 is commented out, but would read a required dataset. I could not find on zenodo the required file, so could not run line 35 in any case. It seems, therefore, that the submitted code cannot be run, at least for this analysis. And a suggestion: instead of writing “Use at your own risk” write something like “Although the code is guaranteed to appropriately and accurately analyse our experimental data to produce the results presented in the published report, any reuse or adaptation of this code for other situations is at own risk.” and put this in each of the code files.

Referee: 3

Comments to the Author(s)

This is a wonderful piece of work addressing a set of compelling questions that tie together theory about evolution and plasticity with understanding the pace and direction of trait responses to climate change in a group of organisms central to ecosystem function.

There are a few minor issues to address.

Line 67-72. Careful with terminology here.

I think genetic adaptation (e.g. via allele frequency change) is normally associated with polymorphism - the existence of different genetically based phenotypes. Plasticity, where the genotypes possess alternative phenotypes, is, I think, polyphenism. It feels like this section doesn't distinguish the mechanism and processes very well and mixes and matches the concepts and terms.

Line 93-94 - this is the first instance of a wider issue about the inclusion of the whole genome re-sequencing data. These data lack the theoretical/conceptual motivation of the rest of the work - even here, the motivation is to link the classy experimental findings with genomic data. This isn't a motivation reflected by the rest of the work where theory underpins expected pattern in specific types of data. As such the genomic data presented don't test any hypotheses and are disconnected from the main questions about evolution and plasticity. It either needs much more development and predictions about what might be expected (e.g. single genes/many genes, classes of genes that might be enriched, links to plasticity vs. allele frequency change) or removal.

Line 140 (Plasticity) - This describes the measurements made, but not the reaction norm. It's clear that the growth rate thermal performance curves are reaction norms and plasticity. What are the 'genotypes' here and the environmental gradient? Needs some additional clarification about where the slope used to define plasticity comes from and motivation. Comparing ancestor to evolved (which this starts with) doesn't align with classic definitions of plasticity arising from variable environments.

Line 150 - This is rather sparse description of the genomic data pipeline. It's also not clear what the endpoints are, why they are being estimated and how key questions are answered (see above)

Line 181 (Plasticity) - lacks details to understand which 'above' method the MS is referring to.

Line 191 - why are the genomic data stats with that section and not in the stats section?

Line 200 - Is this pattern also true in non-microbes? Any evidence to generalise the patterns a bit more?

Line 217 - This feels like one of the most significant results. Would the MS benefit from a more precise definition of population here? Is the phenotypic description? Is this result worth isolating so that the investigation of plasticity vs polymorphism stands out a bit more?

Line 229 - 234 - The bet-hedging theory is a good template. Is it also worth discussion theory about polymorphisms relative to the grain of the environment (the classic literature).

Distinguishing how the pattern is more or less associated with each mechanism associated with how polymorphisms can arise would be valuable. Perhaps introducing more clearly the idea of adaptive tracking, how it is affected by the pace and grain of environment variation and then how the data in the MS on allele frequencies and plasticity fit in might be good.

Line 264 - It would be nice to have a bit more insight in to the arguments about why these expected trade-offs might be relaxed, whether the trade-offs exist in a different biological space, and what the conundrum is if there is no detectable trade-off.

Line 278-280 - This description would be valuable in the methods.

Line 291 - 306 - There are a lot of methods here. Not clear why these are not in the methods..... (see above)

Line 311 - This is a really compelling and interesting result. There is a lot of old theory - Moran 1992 - The evolutionary maintenance of alternative phenotypes; Lively's work; West Eberhard - that might be worth revisiting here?

Line 336 - Is it worth separating out the issue of costs? It's been such an industry that these results deserve a bit of a focus? Does this result also tie into to the lack of trade-offs?

Line 347 - As above - these data and results 'dangle' from the rest of the MS because a firm connection to theory is missing. Can these data be tied more closely to the phenotypic work above and theory so that patterns of relatedness based on SNPS, and the enrichment exercise are more informative? One might also have expected some insight from allele frequency data?

One solution is just to leave these data out. If not, more substantial motivation and theory is needed.

Author's Response to Decision Letter for (RSPB-2021-1418.R0)

See Appendix A.

RSPB-2022-0834.R0

Review form: Reviewer 2

Recommendation

Reject – article is scientifically unsound

Scientific importance: Is the manuscript an original and important contribution to its field?

Good

General interest: Is the paper of sufficient general interest?

Good

Quality of the paper: Is the overall quality of the paper suitable?

Marginal

Is the length of the paper justified?

No

Should the paper be seen by a specialist statistical reviewer?

No

Do you have any concerns about statistical analyses in this paper? If so, please specify them explicitly in your report.

No

It is a condition of publication that authors make their supporting data, code and materials available - either as supplementary material or hosted in an external repository. Please rate, if applicable, the supporting data on the following criteria.

Is it accessible?

N/A

Is it clear?

N/A

Is it adequate?

N/A

Do you have any ethical concerns with this paper?

No

Comments to the Author

I view the revisions and responses as inadequate to be able to recommend publication.

For example:

The response text includes: "We agree that without further in-depth analysis, the genomic investigations have little bearing on the rest of the results (see also reviewer 3) and have hence moved them to the discussion/conclusion section. The photochemical traits, on the other hand, are integral to our question as to whether the amount of phenotypic plasticity that evolves in labile traits differs between environments characterised by short vs long fluctuations. We have rephrased the final paragraph of the introduction (now from line 87), and altered a paragraph the introduction (e.g. from line 72, 81) in order to better tie the introduction to the results/discussion section." (i) Moving irrelevant results to the discussion seems rather a strange approach. (ii) The final paragraph of the introduction begins at line 96 (in the revised ms), so I am not sure what text is being referred to. (iii) In any case I could not find mention in the introduction of photochemical traits or of trait lability. I.e. the introduction does not adequately introduce the study.

The first paragraph of the introduction points out that models assume a trade-off between peak height and curve breadth, and then states that this may be an inappropriate assumption for microbial populations. I.e. "ample scope for the evolution of higher growth rates at elevated T_{opt} and broader curves, or elevated peak rates at higher temperatures with no or negligible effects on curve breadth, i.e. no trade-off between height and breadth [7,8]." The text goes on to state that the AUC is a function of both curve breadth and height, and therefore it is used in the current study. It unclear why the text points out the trade-off assumption and questions it, while the study does not itself address this (or that it is unclear how it does). I would expect analyses of the relationship between height and breadth. Again, I see that the introduction does not adequately introduce the study.

It seems questionable whether AUC can be said to describe changes in "shape". Shape can affect AUC, but so can other things, such as height. Therefore changes in AUC can occur without any change in shape. I think this is pretty fundamental and perhaps a misunderstanding on my part, but the ms text should be able to prevent this.

What is a *valid* evolutionary strategy? I am not familiar with that term.

Review form: Reviewer 3

Recommendation

Accept as is

Scientific importance: Is the manuscript an original and important contribution to its field?

Excellent

General interest: Is the paper of sufficient general interest?

Excellent

Quality of the paper: Is the overall quality of the paper suitable?

Excellent

Is the length of the paper justified?

Yes

Should the paper be seen by a specialist statistical reviewer?

No

Do you have any concerns about statistical analyses in this paper? If so, please specify them explicitly in your report.

No

It is a condition of publication that authors make their supporting data, code and materials available - either as supplementary material or hosted in an external repository. Please rate, if applicable, the supporting data on the following criteria.

Is it accessible?

No

Is it clear?

No

Is it adequate?

No

Do you have any ethical concerns with this paper?

No

Comments to the Author

Thank you for the care and attention to the detail and intent of the referee comments. There were many places to manage the context, history of methods and theory, and semantics arising from all of our reviews. The MS now reflects this effort, strengthening the impact of very insightful results that emerge from excellent experiments driven by this theory. Well done. I think you made a shrewd decision about the molecular data at this point. I look forward to insights that might arise from these data in the future.

Decision letter (RSPB-2022-0834.R0)

16-Jun-2022

Dear Dr Schaum:

Your manuscript has now been peer reviewed and the reviews have been assessed by an Associate Editor. The reviewers' comments (not including confidential comments to the Editor) and the comments from the Associate Editor are included at the end of this email for your reference. As you will see, one of the reviewers has raised some concerns with your manuscript and we would like to invite you to revise your manuscript to address them.

We do not allow multiple rounds of revision so we urge you to make every effort to fully address all of the comments at this stage. If deemed necessary by the Associate Editor, your manuscript will be sent back to one or more of the original reviewers for assessment. If the original reviewers

are not available we may invite new reviewers. Please note that we cannot guarantee eventual acceptance of your manuscript at this stage.

When submitting your revision please upload a file under "Response to Referees" - in the "File Upload" section. This should document, point by point, how you have responded to the reviewers' and Editors' comments, and the adjustments you have made to the manuscript. We also require a copy of the revised manuscript showing track changes to be uploaded.

Research ethics:

Use of animals and field studies:

It is a condition of publication that data supporting your paper are made available either in the electronic supplementary material. Authors must complete the 'data accessibility' section in the submission system. This should list the database and accession number for all data from the article that has been made publicly available, for instance:

NB. From April 1 2013, peer reviewed articles based on research funded wholly or partly by RCUK must include, if applicable, a statement on how the underlying research materials - such as data, samples or models - can be accessed.

[http://datadryad.org/submit?journalID=RSPB&manu=\(Document not available\)](http://datadryad.org/submit?journalID=RSPB&manu=(Document not available)) which will take you to your unique entry in the Dryad repository. If you have already submitted your data to dryad you can make any necessary revisions to your dataset by following the above link.

Please include the Dryad DOI in the Data Accessibility section and reference in the paper's bibliography.

Please see our Data Sharing Policies (<https://royalsociety.org/journals/authors/author-guidelines/>).

Please submit a copy of your revised paper within three weeks. If we do not hear from you within this time your manuscript will be rejected. If you are unable to meet this deadline please let us know as soon as possible, as we may be able to grant a short extension.

Thank you for submitting your manuscript to *Proceedings B*; we look forward to receiving your revision. If you have any questions at all, please do not hesitate to get in touch.

Best wishes,
Professor Hans Heesterbeek
mailto:proceedingsb@royalsociety.org

Associate Editor

Comments to Author:

Your resubmission has been assessed by two of the original reviewers. As you will see their views differ on whether the revisions fully address the issues raised in the original round of review: reviewer 2 is entirely satisfied, whereas reviewer 1 raises some persistent concerns. While I share some of reviewer 1's concerns regarding the need to fully introduce all relevant concepts in the introduction and being transparent about what metrics really measure, I do not believe that these issues are terminal and think that a final revision of the text will be sufficient to address these concerns. As such, please revise the manuscript accordingly.

Reviewer(s)' Comments to Author:

Referee: 2

Comments to the Author(s).

I view the revisions and responses as inadequate to be able to recommend publication.

For example:

The response text includes: "We agree that without further in-depth analysis, the genomic investigations have little bearing on the rest of the results (see also reviewer 3) and have hence moved them to the discussion/conclusion section. The photochemical traits, on the other hand, are integral to our question as to whether the amount of phenotypic plasticity that evolves in labile traits differs between environments characterised by short vs long fluctuations. We have rephrased the final paragraph of the introduction (now from line 87), and altered a paragraph the introduction (e.g. from line 72, 81) in order to better tie the introduction to the results/discussion section." (i) Moving irrelevant results to the discussion seems rather a strange approach. (ii) The final paragraph of the introduction begins at line 96 (in the revised ms), so I am not sure what text is being referred to. (iii) In any case I could not find mention in the introduction of photochemical traits or of trait lability. I.e. the introduction does not adequately introduce the study.

The first paragraph of the introduction points out that models assume a trade-off between peak height and curve breadth, and then states that this may be an inappropriate assumption for

microbial populations. I.e. “ample scope for the evolution of higher growth rates at elevated T_{opt} and broader curves, or elevated peak rates at higher temperatures with no or negligible effects on curve breadth, i.e. no trade-off between height and breadth [7,8].” The text goes on to state that the AUC is a function of both curve breadth and height, and therefore it is used in the current study. It unclear why the text points out the trade-off assumption and questions it, while the study does not itself address this (or that it is unclear how it does). I would expect analyses of the relationship between height and breadth. Again, I see that the introduction does not adequately introduce the study.

It seems questionable whether AUC can be said to describe changes in "shape". Shape can affect AUC, but so can other things, such as height. Therefore changes in AUC can occur without any change in shape. I think this is pretty fundamental and perhaps a misunderstanding on my part, but the ms text should be able to prevent this.

What is a *valid* evolutionary strategy? I am not familiar with that term.

Referee: 3

Comments to the Author(s).

Thank you for the care and attention to the detail and intent of the referee comments. There were many places to manage the context, history of methods and theory, and semantics arising from all of our reviews. The MS now reflects this effort, strengthening the impact of very insightful results that emerge from excellent experiments driven by this theory. Well done. I think you made a shrewd decision about the molecular data at this point. I look forward to insights that might arise from these data in the future.

Author's Response to Decision Letter for (RSPB-2022-0834.R0)

See Appendix B.

Decision letter (RSPB-2022-0834.R1)

08-Jul-2022

Dear Dr Schaum

I am pleased to inform you that your manuscript entitled "Rapid evolution of thermal tolerance and phenotypic plasticity under rapid and slow temperature fluctuations" has been accepted for publication in Proceedings B.

Data Accessibility section

Open Access

Your article has been estimated as being 11 pages long. Our Production Office will be able to confirm the exact length at proof stage.

Paper charges

Sincerely,

Professor Hans Heesterbeek

Associate Editor:

Board Member

Comments to Author:

(There are no comments.)

Appendix A

Response to reviewers PROC B resubmission

Dear editors, dear reviewers,

Thank you for providing such detailed feedback on our manuscript at a time where everyone is being pulled into too many directions. The comments and advice have greatly improved the manuscript. Briefly, we now define all terminology much more carefully, tie the hypotheses mentioned in the introduction to the results more closely, and have decided to put less of a focus on the molecular data. Below, we provide a point-by-point response to the reviewers, where comments are in black, and our response in blue and italics.

Yours sincerely,

Elisa Schaum on behalf of all authors

Associate Editor

Board Member: 1

Comments to Author:

Your ms has been assessed by 3 expert reviewers, each of whom has provided a detailed and thoughtful report. All of the reviewers saw value in the experiments. However, although the reviewers' vary in their overall assessments of the suitability of the work for PRSB, they each raise substantive critiques of the work as it is presented. There appear to be some important conceptual issues that require more attention (e.g. whether you are really measuring plasticity of genotypes?). In addition, 2 of the reviewers questioned whether there was sufficient rationale for the genomic work. There are issues throughout in terms of the clarity of the writing and presentation, particularly with respect to quantitative and bioinformatic methods.

Thank you for summarising the reviewers' concerns. We now define very clearly how plasticity is used the context of this paper, and why we confidently report population level responses (rather than those of individual genotypes, although we have data of clones picked from the evolved populations – see our response to reviewer 1).

In order to avoid self-plagiarising, we reference our 2018 publication for details on the analysis of WGS data. The 2018 publication used the same pipeline for data analysis. We

have kept the genomic data in the manuscript, but now use it as a justification to treat the two subsets of FL-evolved biological replicates separately.

Reviewer(s)' Comments to Author:

Referee: 1

Comments to the Author(s)

This manuscript reports the result of a 300-generation experimental evolution study designed to understand how stable and fluctuating temperature environments influence the evolution of plasticity and tolerance curves in the model diatom *Thalassiosira pseudonana*. The main results reported are effects of treatment on the magnitude of the evolutionary response, a split in the response of the replicates in the 30-40 generation fluctuation treatment, an effect of environment on thermal tolerance curves, an effect of treatment on the degree of plasticity shown by photochemical parameters, details about the patterns of molecular evolution demonstrated in the different treatments. Overall, this seems like a worthy and interesting study but I struggled to follow the hypothesis being tested and how it relates to theory that is frequently alluded to and I have some major concerns about the novelty of the work.

Thank you for the detailed feedback. It is very much appreciated and we address all points below.

1. The study is presented as if this is a new study but there seems to be a lot of overlap with the authors previous 2018 paper (Nature Communications, 1–14. (doi:10.1038/s41467-018-03906-5) which already demonstrated an increase in growth rates at or near selection temperatures, an effect of fluctuating selection on evolutionary response, phenotypic change in metabolic traits and elemental composition, rapid evolutionary shifts in thermal tolerance curves and the molecular divergence between treatments that underpin these differences. It seems that these are the same results presented differently.

The results present here are not the same results presented differently, as these are two separate studies answering two fundamentally different questions.

*This study in deals with fluctuations that differ in **frequency**, rather than being a comparison of fluctuating vs stable environments, which was the focal point of our 2018 study. The data from the slower fluctuations are entirely novel, and are presented in comparison to those of the faster fluctuations and the stable environments that frame the fluctuations for context. We mention our focus on the differences in the frequencies of fluctuations from line 89. One could, in theory, have published one long manuscript including all selection environments, but we would argue that the two questions – and results - are sufficiently different from each other that they warrant to be investigated separately so as to give each set of results and their theoretical underpinning the appropriate level of attention. A single piece presenting all these different results would be too complex and the individual messages would have been lost.*

2. This manuscript reports the results from a slow fluctuation treatment (30-40 generations) that weren't reported in the previous study. A big focus of this study is supposedly comparing what effect 30-40 generation fluctuation has compared to the 3-4 generation effects (previously reported in the 2018 paper). The paper talks about long and short fluctuations leading to different predictions for plasticity, tolerance curves and AUC's but I was a bit confused as to how these predictions were reached and how they would then be directly tested.

Our predictions are based on the existing body of theory as referenced throughout. We test them by measuring phenotypic plasticity of photochemical traits, as well as the AUCs of the tolerance curves for growth under slow and fast fluctuations. We have restructured the introduction to show more clearly why we are comparing environments that differ in the frequencies of their fluctuations. We now also mention explicitly in the introduction that (slowly) fluctuating environments may allow for the evolution of separate strategies, so as to better be able to justify our in-depth discussion of the two population that evolved under the slow fluctuations.

The results don't stick closely to testing these predictions and much of the paper refers to a split in the replicates from the long-fluctuations treatment but no sensible rationale for analysing the data as a split result is given (although it is clear that is what has happened).

We have rephrased parts of the manuscript and the subheaders of the results/discussion section to better link the predictions mentioned in the introduction to the results and discussion. This included (see above) better introducing the idea that a fluctuating environment may allow for more than one valid evolutionary strategy.

The decision to analyse the data as a split result emerged from the data itself – we were not 'planning' this split, but watching the populations as they evolved when the split emerged. One could justify our decision to perform the analyses on the 'split' populations by testing where replicates cluster on a PCoA or similar plot when there is no a priori assumption of a split in the FL treatment. We now state this in the methods and results (line 188, line 250, Supporting Figures 9 and 10). Should the reviewers think it is necessary for better understanding the data, we are happy to also provide an analysis of the data as 'not split' in the supporting information. However, we think that this split is an exciting finding, and have therefore not changed how we present and analyse the data in the main text.

3. The manuscript suggest that theory explains the link between plasticity and evolutionary potential but then doesn't explain what that link is clearly or how we should expect stable and fluctuating environments of different duration (relative to generation time) to influence it.

We now explain more clearly what we mean by 'linking' plasticity in physiology traits and changes in fitness in variable environments from line 82 . There is a vast backdrop of evolutionary theory on plasticity and adaptation, but here, we explicitly test the links between metabolism and growth. Metabolism is a direct driver of growth rate – therefore, the ability to rapidly, and dynamically, adjust metabolic rates will be an important trait when evolving under variable conditions. (Text in manuscript " We explicitly test the links between metabolic rates (as a proxy for plastic traits) and growth rates (as a proxy for

fitness). Metabolism is a direct driver of growth rate – therefore, the ability to rapidly and dynamically, adjust metabolic rates will be a crucial trait when evolving under variable conditions

”)

To look at the evolution of plasticity you would need to isolate single clones and measure plastic responses in the ancestor (single clone) and then compare those responses with the plastic responses of clonal isolates from the evolved populations. But here the authors are just assessing population level plasticity. In this case, it's not possible to know whether the differences observed derive from plasticity or changes in the frequency of clones in the population in the different environments.

There seem to be two issues here: One, the definition of ‘evolution of plasticity’, and two, that we have tested it on the population level.

Here, we compare plastic responses across treatments at the time of 300 generations of evolution. It is true that if we want to look at how the plastic response itself has changed throughout the selection experiment, we have to compare ancestral plasticity (or plasticity evolved under ‘standard laboratory conditions’ in order to account for the fact that there is evolution to any laboratory conditions, too) to plasticity in the selection environment. We have carefully explained this in the methods (from line 142) .

To address the second point, we have growth rate data for single clones (96 per replicate per treatment, see below), but indeed the focus is on population level plasticity throughout for all metabolic traits. The responses could therefore be a result of either plasticity at a cellular level or metabolic diversity within populations. We now state this in line 159 (“Plastic responses were measured on a population level, and therefore, responses could either be due to higher plasticity on a cellular level, or increased metabolic diversity within one sample.

While ideally one should measure plastic responses on the level of single clones isolated from the evolved samples, this is not possible for the methods chosen here. We did, however, re-

clonalise all evolved samples (96 clones per sample, see Supporting Figure 6) and found that within replicate variation for growth was extremely low.”) and line 333 (“We report data on the population level as it is almost impossible to measure these traits on the level of a single cell”).

Response to reviewers Figure 1: *Growth rates of single clones (96 per biological replicate per treatment) isolated at the end of the selection experiment, assayed at 22°C, 26°C, 32°C and under rapid fluctuations. (this was Supporting Figure 6 in the original submission)*

4. I couldn't follow how you generate estimate of plasticity in photosynthesis. A figure might help to explain how you get a measure of photosynthesis in each environment from fitting NP data to a dynamic model of photoinhibition via non-linear least squares regression.

We have added a conceptual/example figure to the SI (now Figure S11, see also below) and refer to it in the methods. There, we show that PS rates at “ I_{opt} ” were extracted from model

fits of oxygen consumption or 'evolution' rates across a light gradient. A similar approach was taken for data returned from the FRRF, where the parameters were also extracted from curves using the appropriate type of non linear least squares regression. We have added the full curves to the supporting information, too.

Response to reviewers Figure 2: *Flow-chart of how data were processed to obtain a measure of phenotypic trait plasticity.*

5. There isn't ever really any attempt to justify the treatments used in relation to natural environments. Given that the experiment started with a single clone from a lab culture it will presumably have adapted to that static 22oC environment. The possible consequences of this should be discussed in the paper. Is not possible to isolate clones from the wild?

This study never intended to replicate the ocean. Rather, we specifically aimed to test in concept how fluctuating environments impact thermal adaptation. While fluctuations are relevant in natural environments (e.g. on daily or seasonal time scales, or during mesoscale mixing), we did not aim to make direct predictions about what happens in nature here, but to show in concept, using a controlled environment, the importance of the factor we are varying (temperature and fluctuation frequencies).

It is indeed possible - and an important next step - to isolate clones from the wild (for example, from environments that differ naturally in the frequencies of their fluctuations), but

this comes with considerable effort, often beyond what is possible during a short contract. From taking the samples to having a clean clone that copes well enough in a laboratory environment to be studied, one might easily spend half a year or more. Here, working on a strain from a culture collection also gave us the advantage of having a known genome to work with, and a known 'evolutionary history' that was not confounded by environmental variability at the sampling location.

Minor comments

Ln 39-41 – logic not clear to me

Most of the introduction has been rephrased – we hope that our arguments now follow logically throughout.

Ln 49-50 – not sure what you mean here? How is evolutionary flexibility got anything to do with being a good descriptor for thermal tolerance?

We meant to say that if the area underneath the curve changes on evolutionary timescales, we can use this measure as a composite of how the shape of the thermal tolerance curve changes. We have rephrased this sentence to “The area underneath the thermal tolerance curve (AUC) is a function of both curve breadth and curve height, and can therefore describe changes in the shape of thermal tolerance curves on short-term and evolutionary timescales”.

Ln 56-59 – this is all too vague and doesn't explain to the reader what hypothesis needs testing.

Rephrased to “Hypotheses concerning under what kind of fluctuations plasticity evolves, and the consequences of changes in plasticity on the shapes of thermal tolerance curves, have yet to be tested empirically for thermal variability (but see a small number of empirical studies for other environmental drivers [21-23]).”, “We explicitly test the links between metabolic rates (as a proxy for plastic traits) and growth rates (as a proxy for fitness). Metabolism is a

direct driver of growth rate – therefore, the ability to rapidly and dynamically, adjust metabolic rates will be a crucial trait when evolving under variable conditions

”and “We tested the magnitudes of the samples’ evolutionary responses, measured whether we found repeatable and predictable changes in AUCs depending on the frequencies of fluctuations, and tested whether plasticity changed in the underlying metabolic traits in environments that fluctuated on shorter vs. longer frequencies”

Ln 67-72 – these are quite clear predictions but not clearly justified. The results section doesn’t stick very closely to what is put forward here.

Thank you for pointing this out. We now explicitly mention these hypotheses again in the results/discussion (see e.g. line 434, 476 ff).

Ln 86 – you don’t give a reference to your previous study here, you need to.

Changed as suggested

Ln 308-309 – adaptive plasticity isn’t expected to evolve in unpredictable environments.

Rephrased to “rapid fluctuations” and removed the part of the sentence referring to environmental predictability, as the fluctuations used here are predictable in both frequency and amplitude, and we cannot make statements about what would happen had they been unpredictable.

Ln 323 – is there a reference to support your statement?

Statement has been rephrased to show it is a general statement rather than a reference to existing literature (now “The area underneath the growth tolerance curve (AUC) describes changes in the shape of the thermal tolerance curve, and the AUC value can be used as a composite measure of the fitness-environment interaction”).

Ln 344-345 – Given that you haven’t previously explained the difference between a cost of plasticity and a limit of plasticity this sentence may confuse readers.

The entire section has been rephrased in a way that we hope addresses the concerns.

Ln349 – 351 – Not clear

Paragraph about molecular evolution has been removed

Fig. 1B – is this the wrong figure?

We agree that the small insert is confusing and have relabelled as A for the large upper plot, B for the small upper plot, C for the large lower plot

Fig. 4 – these correlations don't always appear to hold within treatments. Is it really appropriate to propose a general relationship that is largely a result of treatment effects?

It holds for the samples that have evolved the highest degree of plasticity. Overall, given the data range at treatment level, it would be difficult to detect statistical effects within treatment.

Referee: 2

Comments to the Author(s)

In the time I had available for this review I was not able to work through the entire manuscript and supplementary information. I apologise that I could not devote more time to the task.

The submitted material concerns what appears to be a well-motivated experimental study of evolution of thermal performance curves in constant and fluctuating temperature environments. The subject matter is, in my opinion, of general interest to ecologists and evolutionary biologists, and is therefore likely of interest to the general audience of PRSB.

Nevertheless, there are many things that can and need to be addressed. The standard of presentation of the research appears to be insufficient for publication. Important elements of the experimental design are not given. Quantitative methods are unclear, and some seem inappropriate. Not all the material presented seems necessary given the Introduction. Some

results stated in the text seem to be contradicted by results presented in the figures and analysis tables.

I give below details of some of these issues, and of other issues, but please recall that I could not finish working through the entire ms in the time I had available. Therefore please extrapolate my findings to the conclusion that all aspects of the submitted material should be thoroughly and critically reviewed and revised before any resubmission.

Point-by-point comments follow:

Thank you for having taken time to review this manuscript despite being pressed for time.

Abstract line 19: please consider stating how the frequency determines the shape of the tolerance curve, rather than just that it does.

As we are very close to the word limit as is we have not changed the abstract, but the introduction has been reworded throughout in order to more clearly show the relationship between environmental fluctuations and the shape of tolerance curves.

Line 38. It is unclear what the “this” refers to, since the contents of the previous sentence do not clearly create the implication of constant area. And please consider referencing evidence for the statement.

Rephrased and referenced

Line 42. Please be more specific about what are “these assumptions”. Currently a reader has to try to sort them out from the previous text, which is not ideal.

Rephrased to “the assumption of a trade-off”

Line 86, please consider referencing evidence for the statement.

Referenced as suggested

The genomic investigation appears to have little or no conceptual basis presented in the Introduction. Same goes for the photochemical traits work. This is a theme that recurs: e.g. tracking and analysing fitness trajectories is not well justified in the Introduction. More

generally, it seems that the submitted manuscript is more of a report of everything that was done in project, compared to a reporting of only the results required to address the matters raised in the Introduction. A critical review of content seems appropriate.

We agree that without further in-depth analysis, the genomic investigations have little bearing on the rest of the results (see also reviewer 3) and have hence moved them to the discussion/conclusion section. The photochemical traits, on the other hand, are integral to our question as to whether the amount of phenotypic plasticity that evolves in labile traits differs between environments characterised by short vs long fluctuations. We have rephrased the final paragraph of the introduction (now from line 87), and altered a paragraph the introduction (e.g. from line 72, 81) in order to better tie the introduction to the results/discussion section.

The nature of the “cycling” between 22 and 32 is unclear. Was it a sinusoidal cycle? Or was it switching between the two temperatures? This seems important, as a square-wave type fluctuation seems much less relevant than a smooth one. Also, how were treatments and replicates distributed among incubators (particularly the three FL1 and three FL2 replicates)?

It was indeed a (manually moving between 22 °C and 32 °C incubators) ‘switching’ rather than a smooth transition. All FL samples were treated the same - we can be sure that handling effects do not explain the differences between the replicates. It is true that a square-wave type fluctuation has little semblance of the fluctuations organisms would experience in a ‘real’ ocean, but as pointed out in our response to reviewer 1, this study is a necessarily conceptual first step on route to a more ‘realistic’ understanding of more complex scenarios. Additionally, while the transfer between the incubators will have been quick, the seawater in the flasks would not have immediately changed temperature, making it likely that the samples would have ‘seen’ the change in temperature as a more sinusoidal cycle.

We have replaced ‘cycling’ with ‘switching’ throughout.

Line 182 states “Phenotypic plasticity was calculated as described above...”. I’m sorry if I

missed something obvious, but I don't see where this was described "above".

It referred to our explanation of how plasticity was calculated for photosynthesis rates. We have rephrased both the section about photosynthesis rates and section the reviewer refers to here (now from line 133, and line 142).

There is considerable duplication of text between some sections of the main text and the supplementary information. It then takes more time than it should to find any unique information (e.g. about the nature of the fluctuations).

Done where applicable

Line 171. Why not ln-normalised?

We used a GAMM fit, which unlike other thermal tolerance fits (e.g. the Sharpe-Schoolfield), does not require that the data be ln-normalised across a temperature gradient.

Figure 1A and inset. Please display all data points. And there is little point in having the different colours; it might even be confusing to have them. And on the version of the figure with the caption below there is no numeric scale or dashed line at $y=1$ on the inset. And please help the reader translate between generations and days. E.g. put number of generations on the x-axis, rather than number of days.

We have decided to keep time in days rather than generations as all samples would have reached slightly different generations on particular days. We have added ~100, ~200, and ~300 generations to the figure. The figure is now also in png format as we assume that the PDF conversion deleted the dashed line, data points, and numeric scale. We have left in the colour scheme as it re-occurs throughout the manuscript (but will remove it should it be necessary).

Analysis of the evolutionary response (Table 1). The sentence starting "We fitted a..." implies that growth rate of the evolved sample was the response variable. The next sentence, however, suggests the response variable might have been a relative growth rate. Thus I am already confused. The description goes on to state that a full model with selection

environment as a fixed factor and biological replicate as a random factor was used. This implies that there were repeated measures, i.e. multiple measures for each biological replicate. But this doesn't correspond with the context of Figure 1A, where it is only the terminal response. Finally, why was model selection performed at all? There is only one fixed factor (treatment)... why not just use an F-test (or likelihood ratio test) on the treatment. Why was model selection used in each of the analyses it was used in?

We have rephrased the section (line175 "To compare the fold-changes of evolved samples (after 300 generations 't300') compared to the growth rate of the ancestor at the respective selection temperature, the ratios of growth of the ancestor at selection temperature, and the evolved sample at selection temperature were calculated, and a linear mixed model used, where that ratio was the response variable ")

It is true that we are not dealing with a repeated measure at the time point of calculating the response, as we are referring to the end point here. We still need to fit biological replicate as a random factor here due to the nature of the replicates (they are all related to each other as they date back to the same ancestral clone). We used model selection to also test whether treatment effects were larger than those of the biological variation between replicates.

Line 238-244 content would be better placed in the Methods, or even Introduction.

A similar statement is already in the introduction – we think it makes sense to repeat the sentiment here to better tie the introduction to the results.

Line 247 states "Populations evolved in the fluctuating environments had faster growth at temperatures exceeding 32°C (Fig 2 A, Supporting Tables 3 and 6) than samples evolved in stable conditions." But Figure 2A and 2D shows that the FL2 replicates did not have higher growth rate above 32°C than samples evolved in stable conditions. That is, the result stated in the text is not supported by the data presented in the figures. This discrepancy also appears to occur in the following two paragraphs, one about AUC and one about optimum temperature.

Thank you for pointing this out. We have rephrased this to “Populations evolved in the rapidly fluctuating environments had faster growth at temperatures exceeding 32°C (Fig 2 A, Supporting Tables 3 and 6) than samples evolved in stable conditions. Half of the replicates from the environment that fluctuated on longer time scales (FL) behaved similarly to those evolved under rapid fluctuations. The thermal tolerance curves of the remaining FL- evolved samples more closely resembled tolerance curves of samples evolved under stable conditions.”

Finally, I took a quick look at the zenodo code and data submission, and tried to run the 20210610_Thally02_Figure02_plot_and_stats.R script.... Line 35 is commented out, but would read a required dataset. I could not find on zenodo the required file, so could not run line 35 in any case. It seems, therefore, that the submitted code cannot be run, at least for this analysis. And a suggestion: instead of writing “Use at your own risk” write something like “Although the code is guaranteed to appropriately and accurately analyse our experimental data to produce the results presented in the published report, any reuse or adaptation of this code for other situations is at own risk.” and put this in each of the code files.

Thank you for pointing this out – we hope the issue is fixed now! The file you mentioned is NOT required to run the analysis for the data presented as it is an in-depth analysis at t100, whereas here we focus on t300. We have removed all t100 analyses from the file altogether to avoid such confusion. We have also opted for a more scientific way of introducing the code, and have removed the colloquialisms.

Referee: 3

Comments to the Author(s)

This is a wonderful piece of work addressing a set of compelling questions that tie together theory about evolution and plasticity with understanding the pace and direction of trait responses to climate change in a group of organisms central to ecosystem function.

There are a few minor issues to address.

Thank you very much for the positive feedback!

Line 67-72. Careful with terminology here.

I think genetic adaptation (e.g. via allele frequency change) is normally associated with polymorphism - the existence of different genetically based phenotypes. Plasticity, where the genotypes possess alternative phenotypes, is, I think, polyphenism. It feels like this section doesn't distinguish the mechanism and processes very well and mixes and matches the concepts and terms.

Rephrased and terminology explained better (we hope).

Line 93-94 - this is the first instance of a wider issue about the inclusion of the whole genome re-sequencing data. These data lack the theoretical/conceptual motivation of the rest of the work - even here, the motivation is to link the classic experimental findings with genomic data. This isn't a motivation reflected by the rest of the work where theory underpins expected pattern in specific types of data. As such the genomic data presented don't test any hypotheses and are disconnected from the main questions about evolution and plasticity. It either needs much more development and predictions about what might be expected (e.g. single genes/many genes, classes of genes that might be enriched, links to plasticity vs. allele frequency change) or removal.

We agree and have removed the molecular data from the results/discussion. We do use the SNV data to strengthen our point of the differences in the FL-evolved replicates.

Line 140 (Plasticity) - This describes the measurements made, but not the reaction norm. It's clear that the growth rate thermal performance curves are reaction norms and plasticity. What are the 'genotypes' here and the environmental gradient? Needs some additional clarification about where the slope used to define plasticity comes from and

motivation. Comparing ancestor to evolved (which this starts with) doesn't align with classic definitions of plasticity arising from variable environments.

We have rephrased, and (also in response to reviewer 1) added a flow chart to the supporting information (Supporting Figure 11) that shows we measure the reaction norms of evolved samples to test for plasticity in evolved samples. We compare this to the reaction norms of the ancestor to test to which degree plasticity has evolved.

Line 150 - This is rather sparse description of the genomic data pipeline. It's also not clear what the endpoints are, why they are being estimated and how key questions are answered (see above)

We have removed most of the genomic data from the results/discussion – and think that now pointing the reader to our 2018 pipeline will suffice.

Line 181 (Plasticity) - lacks details to understand which 'above' method the MS is referring to.

Clarified as suggested (now from line 142)

Line 191 - why are the genomic data stats with that section and not in the stats section?

Genomic data have been largely removed

Line 200 - Is this pattern also true in non-microbes? Any evidence to generalise the patterns a bit more?

This may be rather tricky – in non-microbes with distinct developmental stages, the timing of fluctuations and the matching (or mismatching) of the associated plastic phenotypes is a much more complex issue and we dare not speculate.

Line 217 - This feels like one of the most significant results. Would the MS benefit from a more precise definition of population here? Is the phenotypic description? Is this result worth isolating so that the investigation of plasticity vs polymorphism stands out a bit more?

Rephrased to make clear that a 'population' refers to a biological replicate here, rather than separate phenotypes co-existing within the same culture flasks.

Line 229 - 234 - The bet-hedging theory is a good template. Is it also worth discussion theory about polymorphisms relative to the grain of the environment (the classic literature). Distinguishing how the pattern is more or less associated with each mechanism associated with how polymorphisms can arise would be valuable. Perhaps introducing more clearly the idea of adaptive tracking, how it is affected by the pace and grain of environment variation and then how the data in the MS on allele frequencies and plasticity fit in might be good.

Thank you for this suggestion. We added a few sentences throughout the results/discussion to introduce the idea of adaptive tracking (and pace/grain of the environment) more clearly.

Line 264 - It would be nice to have a bit more insight in to the arguments about why these expected trade-offs might be relaxed, whether the trade-offs exist in a different biological space, and what the conundrum is if there is no detectable trade-off.

We cannot say for sure why plasticity is not very costly in our system though we argue that there is the potential for costs existing in traits and/or environments not measured here, e.g. during sexual rather than clonal reproduction. We have also added a sentence stating that the conundrum largely lies in plasticity being potentially overestimated in most current models.

Line 278-280 - This description would be valuable in the methods.

Line 291 - 306 - There are a lot of methods here. Not clear why these are not in the methods..... (see above)

Both have been added to the methods. We reiterate some of the main statements in the main text because we think that not everyone might read the methods.

Line 311 - This is a really compelling and interesting result. There is a lot of old theory
- Moran 1992 - The evolutionary maintenance of alternative phenotypes; Lively's work;
West Eberhard - that might be worth revisiting here?

We have elaborated on some of our statements and added references as suggested, although most of the references describe long-lived organisms with distinct developmental stages, where the time point of the fluctuation in relation to the developmental stage is crucial with regards to the phenotypic outcome.

Line 336 - Is it worth separating out the issue of costs? It's been such an industry that these results deserve a bit of a focus? Does this result also tie into to the lack of trade-offs?

We have moved, and slightly rephrased, this section to give it a bit more focus

Line 347 - As above - these data and results 'dangle' from the rest of the MS because a firm connection to theory is missing. Can these data be tied more closely to the phenotypic work above and theory so that patterns of relatedness based on SNPS, and the enrichment exercise are more informative? One might also have expected some insight from allele frequency data?

One solution is just to leave these data out. If not, more substantial motivation and theory is needed.

These data have now been moved out

Appendix B

Dear Editors, dear reviewers

Thank you for having taken the time to assess our resubmission. We are delighted to hear that the original reviewer 3 is satisfied with the changes we made, and we address all comments made by the original reviewer 2 below. Reviewer comments are in black; our responses are in blue italics.

Referee: 2

The response text includes: "We agree that without further in-depth analysis, the genomic investigations have little bearing on the rest of the results (see also reviewer 3) and have hence moved them to the discussion/conclusion section. The photochemical traits, on the other hand, are integral to our question as to whether the amount of phenotypic plasticity that evolves in labile traits differs between environments characterised by short vs long fluctuations. We have rephrased the final paragraph of the introduction (now from line 87), and altered a paragraph the introduction (e.g. from line 72, 81) in order to better tie the introduction to the results/discussion section." (i) Moving irrelevant results to the discussion seems rather a strange approach. (ii) The final paragraph of the introduction begins at line 96 (in the revised ms), so I am not sure what text is being referred to. (iii) In any case I could not find mention in the introduction of photochemical traits or of trait lability. I.e. the introduction does not adequately introduce the study.

We apologise for any confusion that might have arisen due to a mismatch of line numbers – this might have been due the fact that we submitted to versions: one track changed, and one ‘clean’ version. We refer to the line numbers in the ‘clean’ document throughout. To address the reviewer’s remarks

- (i) we do not think that the WGS data are irrelevant, and the data are never described as irrelevant in the main manuscript. Rather, we think that in the manuscript’s current state, the data serve to confirm our phenotypic findings, particularly in highlighting the emergence of two strikingly different populations in the “FL” treatment which are distinct both in their phenotypic traits and the WGS analysis.*
- (ii) in this resubmission, the paragraph we meant to refer to now starts in line 67 (but see point iii); we have extensively rewritten the introduction). The paragraph detailing our approach and research questions (“Here, we...”) now starts in line 91*
- (iii) we have now made extensive changes to the introduction (we have not used track changes as we essentially restructured the entire introduction) in order to accommodate the referee’s concerns*

The first paragraph of the introduction points out that models assume a trade-off between peak height and curve breadth, and then states that this may be an inappropriate assumption for microbial populations. I.e. “ample scope for the evolution of higher growth rates at elevated T_{opt} and broader curves, or elevated peak rates at higher temperatures with no or negligible effects on curve breadth, i.e. no trade-off between height and breadth [7,8].” The text goes on to state that the AUC is a function of both curve breadth and height, and therefore it is used in the current study. It unclear why the text points out the trade-off assumption and questions it, while the study does not itself address this (or that it is unclear how it does). I would expect analyses of the relationship

between height and breadth. Again, I see that the introduction does not adequately introduce the study.

It is true that the trade-off or lack thereof are not the main focus of this study. Our point was that since a trade-off is unlikely in microbial organisms, a measure that contains changes in height as well as breadth might be better suited how thermal reaction norms change with temperature on micro evolutionary time scales. As we have extensively reworked the introduction, and included this explanation, we hope to have addressed this concern, too.

It seems questionable whether AUC can be said to describe changes in "shape". Shape can affect AUC, but so can other things, such as height. Therefore changes in AUC can occur without any change in shape. I think this is pretty fundamental and perhaps a misunderstanding on my part, but the ms text should be able to prevent this.

We had considered height to be a 'shape' characteristic and have rephrased accordingly (now line 215, "[...] including height")

What is a *valid* evolutionary strategy? I am not familiar with that term.

We consider an evolutionary strategy that leads to an increase in fitness as 'valid'. However, we have rephrased to simply 'adaptive' as this may be more accessible to the broad readership of the journal.